# Flagellar cAMP signaling controls trypanosome progression through host tissues

Sebastian Shaw [1,2], Stephanie F. DeMarco [3], Ruth Rehmann[1], Tanja Wenzler[1], Francesca Florini[1,2], Isabel Roditi [1] & Kent L. Hill [3,4,5]

The unicellular parasite *Trypanosoma brucei* is transmitted between mammals by tsetse flies. Following the discovery that flagellar phosphodiesterase PDEB1 is required for trypanosomes to move in response to signals in vitro (social motility), we investigated its role in tsetse flies. Here we show that PDEB1 knockout parasites exhibit subtle changes in movement, reminiscent of bacterial chemotaxis mutants. Infecting flies with the knockout, followed by live confocal microscopy of fluorescent parasites within dual-labelled insect tissues, shows that PDEB1 is important for traversal of the peritrophic matrix, which separates the midgut lumen from the ectoperitrophic space. Without PDEB1, parasites are trapped in the lumen and cannot progress through the cycle. This demonstrates that the peritrophic matrix is a barrier that must be actively overcome and that the parasite's flagellar cAMP signaling pathway facilitates this. Migration may depend on perception of chemotactic cues, which could stem from co-infecting parasites and/or the insect host.

[1] Institute of Cell Biology, University of Bern, Baltzerstrasse 4, CH-3012 Bern, Switzerland. [2] Graduate School for Cellular and Biomedical Sciences, University of Bern, CH-3012 Bern, Switzerland. [3] Molecular Biology Institute, University of California, Los Angeles, CA 90095, USA. [4] Department of Microbiology, Immunology and Molecular Genetics, University of California, Los Angeles, CA 90095, USA. [5] California NanoSystems Institute, University of California, Los Angeles, CA 90095, USA. These authors contributed equally: Sebastian Shaw, Stephanie F. DeMarco. These authors jointly supervised: Isabel Roditi, Kent L. Hill. Correspondence and requests for materials should be addressed to I.R. (email: isabel.roditi@izb.unibe.ch) or to K.L.H. (email: kenthill@microbio.ucla.edu)

A common feature of parasitic protozoa is the need to sense and adapt to diverse environments in different hosts and tissues within these hosts. At present, however, little is known about mechanisms of signal transduction in these organisms and how these impact transmission and pathogenesis. *Trypanosoma brucei ssp* are medically and economically important parasites that are prevalent in sub-Saharan Africa. Two subspecies, *T. b. gambiense* and *T. b. rhodesiense* are responsible for human sleeping sickness, while *T. b. brucei* causes the animal disease Nagana. Restriction of the parasites to sub-Saharan Africa is determined by the geographic range of the tsetse fly, which is their definitive host and is crucial for their transmission between mammals.

Like many unicellular parasites, *T. brucei* has a complex life cycle that requires it to undergo several rounds of differentiation, migrate through diverse tissues, and traverse a variety of barriers in both its mammalian and fly hosts[1]. At least two forms exist in the mammal, a proliferative slender form and a quiescent stumpy form that is preadapted for transmission when tsetse flies take a blood meal from an infected animal[2]. Transition between these two developmental forms occurs in response to an extracellular signal[3]. Following ingestion by the fly, the blood meal rapidly passes to the crop, after which it is transferred to the lumen of the posterior midgut (Fig. 1)[4,5]. Here, stumpy forms differentiate into early procyclic forms and replace the mammalian-specific variant surface glycoprotein coat with a mixture of GPEET and EP procyclins[6,7]. To progress further through their life cycle, the parasites must gain access to the ectoperitrophic space. This entails crossing the peritrophic matrix (PM), a trilaminar sheath of chitin, (glyco)proteins, and glycosaminoglycans[8]. At present, the site and mechanism of crossing are unclear[9]. Establishment of midgut infection correlates with parasite differentiation to late procyclic forms, which are EP-positive, but GPEET-negative[7]. As the infection proceeds, parasites fill the ectoperitrophic space and move toward the anterior midgut[10–12]. Two other morphological forms have been described in this compartment, long procyclic forms[12] and mesocyclic forms[1,10].

In the next phase of the life cycle, parasites must cross the PM a second time. This occurs at the proventriculus (or cardia), the junction between the mid- and foregut and site of PM secretion[8]. Although colonization of the proventriculus was described more than a century ago[4], relatively little attention has been paid to the role of this organ in the trypanosome life cycle[10–15]. From the proventriculus, the parasites move via the foregut to the salivary glands. A variety of post-mesocyclic forms have been described, including long epimastigotes that undergo an asymmetric division[10,11] and deliver short epimastigotes to the salivary glands. Short epimastigotes colonize the salivary gland epithelia, completing the cycle with differentiation to metacyclic forms that can be transmitted to a new mammalian host[1].

Throughout its developmental cycle, *T. brucei* must be able to sense its environment and transduce signals that effect its differentiation to the next developmental stage and/or movement to the next compartment. In many organisms, cyclic nucleotides are important second messengers that direct cellular responses to external signals. Well-studied examples include chemotaxis of invertebrate sperm[16], as well as fruiting body formation in *Dictyostelium discoideum*, where cAMP acts as both signal and second messenger[17]. In bacteria, intracellular cyclic nucleotides regulate the transition between biofilm formation and swarming motility in response to quorum-sensing signals[18], thereby impacting both differentiation and movement. In general, cyclic nucleotide levels in these systems are controlled by reciprocal activities of nucleotide cyclases that generate the signal and phosphodiesterases (PDEs) that remove the signal[18].

The *T. brucei* genome encodes cAMP signaling components, although these differ in several respects from those in mammalian cells[19]. For example, the *T. brucei* protein kinase A does not appear to be directly responsive to cAMP[19]. *T. brucei* also lacks conventional G protein-coupled receptors (GPCRs), which typically mediate extracellular ligand-dependent activation of adenylate cyclase (AC)[20]. This deficit seems to be accommodated by a family of ~75 receptor-type ACs[21,22] that are structurally very different than their mammalian counterparts and not affected by pharmacological treatments that activate mammalian ACs. The catalytic domain of trypanosome ACs is connected by a single transmembrane domain to a variable extracellular domain. This architecture offers the potential to regulate cAMP production by external ligands binding directly to the cyclase, rather than to an upstream GPCR. All ACs investigated so far are localized to the flagellum, where they have been implicated in parasite signaling and motility in culture[23–25]. AC isoforms are differentially expressed throughout the *T. brucei* life cycle[21,23,24,26–28], suggesting they may each respond to distinct cues encountered only in specific host tissues. Consistent with a role for cAMP in differentiation, increased AC activity was observed to coincide with stumpy to procyclic differentiation in vitro[29,30]. In bloodstream form *T. brucei*, the bloodstream stage-specific AC, ESAG4, has been demonstrated to influence infection in the mammalian host[31]. In this case, however, cAMP was proposed to act on host cell function rather than as a second messenger within the parasite.

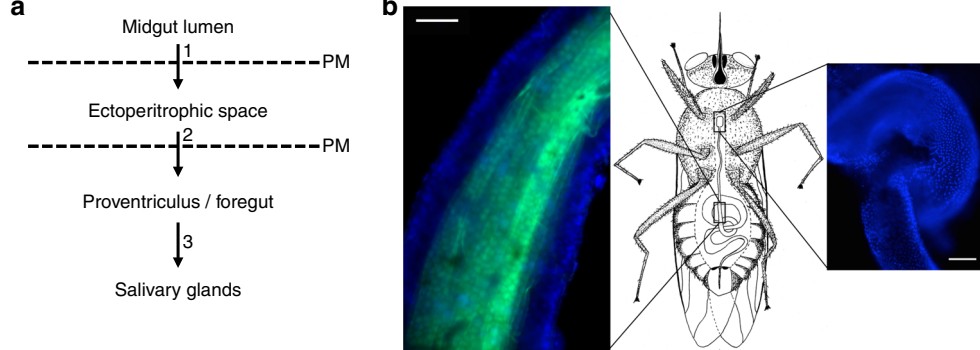

**Fig. 1** Course of migration by trypanosomes and anatomical context in the tsetse fly. **a** Schematic depiction of the path taken by trypanosomes during cyclic transmission, with numbers 1–3 marking major tissue transitions. PM: peritrophic matrix. **b** Schematic of a tsetse fly (central panel), with boxed regions indicating the location of the midgut (left panel) and proventriculus (right panel). Left panel, an isolated tsetse fly midgut in which the nuclei of epithelial cells are stained with Hoechst dye (blue) and the PM is stained with fluorescein-tagged wheat germ agglutinin (green). Right panel, an isolated tsetse fly proventriculus stained with Hoechst dye (blue) to visualize nuclei. Scale bar: 100 microns

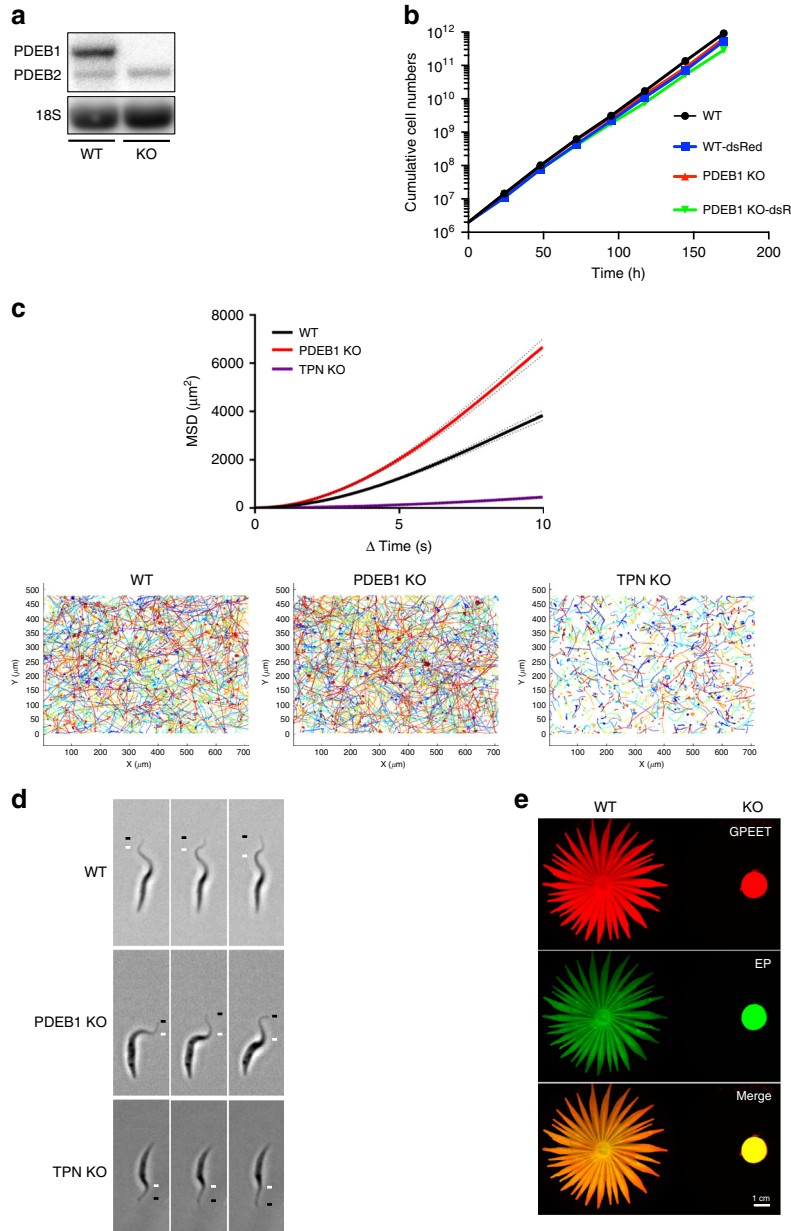

**Fig. 2** Effect of PDEB1 knockout on SoMo, growth and motility. **a** Northern blot analysis of RNA prepared from wild-type parental cells (WT) or PDEB1 knockout cells (KO). Blots were hybridized with a probe corresponding to the coding region of PDEB1. PDEB2 is 93% identical to PDEB1 and the probe cross-hybridizes weakly under the conditions used. 18S rRNA serves as a loading control[66]. **b** Comparison of population doubling times of WT, KO, WT-dsRed, and KO-dsRed cell lines in suspension culture over the course of 7 days. Cell densities were adjusted daily to $3 \times 10^6$ cells ml$^{-1}$ in order to ensure logarithmic growth. **c** Mean-squared displacement (MSD) is measured for WT, PDEB1 KO, and trypanin knockout (TPN KO) cells. Results are from three biological replicates and a total of $n = 1449$ cell traces for WT, $n = 1339$ cell traces for KO, and $n = 1208$ cell traces for TPN KO from a total of 34 videos for each. Dotted lines represent standard error of the mean (SEM). Cell traces from the 34 videos are superimposed over each other for WT, PDEB1 KO, and TPN KO, respectively. **d** Still images from high-speed videos of WT, PDEB1 KO, and TPN KO cells taken at 496 frames per second under x20 magnification. Each series of images shows one cell per genotype at six millisecond intervals. The black line indicates the flagellum tip, and the white line indicates the peak of the flagellar wave. See supplemental information for Supplementary Movies 1–3. **e** Social motility assays were performed with WT and KO cells. Community lifts[36] were performed and incubated with anti-EP (green) and anti-GPEET (red) antibodies

The *T. brucei* genome encodes five PDEs: PDEA, PDEB1, PDEB2, PDEC, and PDED. PDEA is neither essential for bloodstream or procyclic forms, nor is it required for fly midgut infection[32]. RNAi knockdown of PDEB1 and PDEB2 together is lethal in bloodstream form trypanosomes, but knockdown of either protein alone does not affect viability[33] and their role in trypanosome biology, beyond being essential, is unclear. Knockdown of PDEB1 and PDEB2, singly or in combination, does not

have detrimental effects on growth or motility of procyclic forms in liquid culture[33].

Evidence for a role of cAMP beyond parasite viability has come from studies of collective cell motility[25,34]. When early procyclic forms are cultured on a semi-solid surface, they exhibit a type of coordinated group movement termed social motility (SoMo) in which the parasites assemble into groups that sense signals from other cells and alter their movement in response[35,36]. Genetic or

pharmacological inhibition of PDEB1 blocks SoMo, while reduced expression or ablation of catalytic activity of specific ACs results in hypersocial behavior[25,34]. These findings indicate that flux through the cAMP pathway regulates how the parasites respond to signals from their environment. Direct measurement of intracellular cAMP levels in live trypanosomes supports this idea, and experiments with cAMP analogues demonstrate that cAMP itself is the active molecule, rather than metabolic breakdown products[34]. Together, these studies demonstrate that *T. brucei* harbors a functional cAMP signal transduction pathway and that signals from this pathway can alter parasite behavior. To date, however, it is not known whether this pathway is required for infection or transmission.

Here, we infect tsetse flies with a *T. brucei* PDEB1 deletion mutant and use dual labeling of parasites and fly tissues to interrogate the role of flagellar cAMP signaling in parasite movement through the fly. We show that PDEB1 is required for traversal of the peritrophic matrix, a chitinous structure that separates the fly midgut lumen from the midgut epithelium. Without PDEB1, most parasites remain trapped in the midgut lumen and the transmission cycle is aborted. Our results reveal a tissue-specific requirement for cAMP signaling and show that the flagellar cAMP signaling pathway of *T. brucei* is crucial for successful progression through tissues in the tsetse fly host.

## Results

**Generation of a *T. brucei* PDEB1-null mutant**. Previous work showed that phosphodiesterase PDEB1 mRNA was reduced 90% by RNAi in procyclic forms with only minimal effects on growth[34]. Given the residual PDEB1 mRNA, together with the uncertainty of whether knockdown occurs equally efficiently in different fly tissues, we reasoned that these lines would not be suitable for fly infection studies. Therefore, as a prelude to examining the role of PDEB1 during transmission by the tsetse fly, a null mutant of PDEB1 was generated in early procyclic forms. PCR data (Supplementary Figure 1) and northern blot analysis confirmed that the PDEB1 gene was deleted from the knockout (KO), that mRNA was not detectable, and that the level of PDEB2 mRNA was unchanged compared with the parental line (Fig. 2a). Knocking out PDEB1 had no impact on the population doubling time (KO, $9.1 \pm 0.36$ h) compared with the wild-type parent (WT, $9.0 \pm 0.21$ h; Fig. 2b). WT and KO cells were also tagged with cytoplasmic dsRed for later experiments. Expression of dsRed had a slight effect on growth, but population doubling times were not substantially different between WT-dsRed ($9.5 \pm 0.07$ h) and KO-dsRed ($9.9 \pm 0.21$ h; Fig. 2b). Since the environment in the fly is presumed to be glucose-poor, we also tested growth in a medium with or without glucose. Both WT and KO grew equally well irrespective of whether glucose was present or not (Supplementary Figures 2h, i).

Because parasite propulsive motility impacts fly infection[37], we performed motility tracing in suspension culture. We also generated a trypanin knockout (TPN KO) in the same background as PDEB1 KO to serve as a control for a known motility mutant. WT, KO, and TPN KO cells were then assessed for propulsive motility. PDEB1 KO cells do not exhibit a loss of motility; rather, they had a greater mean-squared displacement (MSD) than WT. TPN KO cells have a severe motility defect, as expected (Fig. 2c). Because MSD incorporates processivity, differences in MSD can be due to the speed of a cell and/or how straight it moves. Therefore, we examined curvilinear (VCL) and straight-line velocity (VSL) for each population of WT, KO, and TPN KO cells. The VCL and VSL distribution of cell trajectories indicate that WT and PDEB1 KO cells move similarly, while TPN KO clearly shows reduced VSL (Supplementary

Figure 3). The mean linearity, a ratio of VSL/VCL[38], of PDEB1 KO is similar though slightly larger than WT, while that of TPN KO cells is substantially reduced (Supplementary Figure 3). These results indicate that the greater MSD in PDEB1 KO cells may reflect less turning compared with WT. Note that reduced turning is also observed in bacterial chemotaxis regulatory mutants[39]. High-speed video analysis of WT, KO, and TPN KO cells shows that WT and KO cells exhibit the normal three-dimensional tip-to-base flagellar waveform, while TPN KO cells do not (Fig. 2d and Supplementary Supplementary Movies 1–3).

When analyzed for SoMo (Fig. 2e), the PDEB1 KO showed the same SoMo-negative phenotype as the RNAi line. Since SoMo is restricted to early procyclic forms[40] one possibility was that removal of PDEB1 caused cells to differentiate to late procyclic forms. To test this, expression of the early procyclic form marker GPEET was assessed by community lifts (Fig. 2e) and flow cytometry (Supplementary Figure 2b, d). These analyses showed that 99% of the KO cells were GPEET-positive, indicating they are indeed early procyclic forms.

**PDEB1 knockout has a defect in establishing a fly infection**. To address the role of cAMP signaling in *T. brucei* during infection of its insect vector, teneral (newly hatched) tsetse flies were infected with WT or KO parasites. Flies were dissected at 3, 7, and 14 days post infection, and midguts were scored for the prevalence and intensity of infection. Fly midguts were assessed at day 3 to determine whether parasites are able to survive in the midgut lumen. Between days 3 and 7, trypanosomes cross the PM and enter the ectoperitrophic space[12,41]. Finally, by day 14 parasites should have established a chronic midgut infection and reached the proventriculus[10,12,41]. In comparison with WT-infected flies, flies infected with the KO showed a decreased prevalence and intensity of infection on days 7 and 14 (Fig. 3a). On day 14, the endpoint of the experiment, there was a statistically significant difference in midgut infection rates ($p = 0.0144$, Fisher's exact test, two-sided). At day 14, proventriculi from flies were also examined and a major difference was observed between WT- and KO-infected flies ($p < 0.0001$, Fisher's exact test, two-sided; Fig. 3b). In the WT group, 13 flies with a heavily infected midgut were scored for proventriculus infection; all had a heavily infected proventriculus. However, of 10 flies with a heavy midgut infection in the KO group, only one fly showed a proventriculus infection, and the intensity was weak. Therefore, even if a heavy midgut infection was established, the PDEB1-KO mutant was rarely able to progress to proventriculus infection.

**PDEB1 addback rescues defects in SoMo and infection**. To confirm that the SoMo and fly infection defects of KO cells were due to the absence of PDEB1, the KO was transfected with a copy of PDEB1 that integrates upstream of a procyclin locus. Northern blot analysis indicated that two independent addback clones, AB1 and AB2, expressed PDEB1 at levels 4.3-fold and 4.2-fold higher than WT (Fig. 4a). We noted that the two clones had slightly longer population doubling times (AB1, 10 h; AB2, 10.4 h) than WT or KO parasites (Fig. 4b), which might be due to the overexpression of PDEB1. Motility analysis of AB1 and AB2 showed that both clones have MSD and VSL/VCL ratios close to that of WT (Fig. 4c and Supplementary Figure 3).

The two addback clones were assessed for SoMo. In both cases, the addbacks formed radial projections, although they were fewer in number than WT (Fig. 4d; Supplementary Figure 4). Monitoring expression of GPEET by flow cytometry confirmed that the vast majority of addback cells (99%; Supplementary Figures 2f and g) were still early procyclic forms. Note that the differences observed in SoMo between WT and addbacks are not

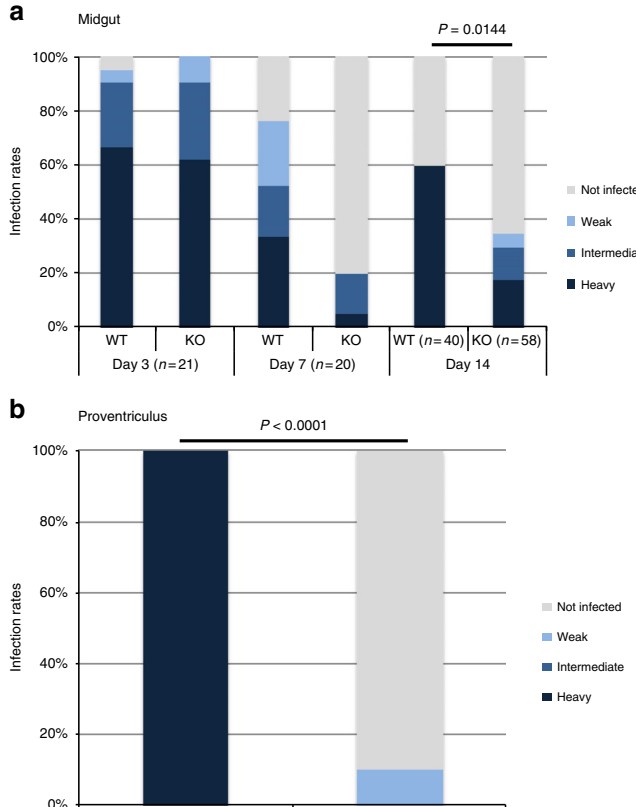

**Fig. 3** PDEB1 is required for colonization of the proventriculus. **a** Prevalence and intensity of midgut infections. Teneral flies were infected at day 0 and dissected to remove the midgut 3, 7, and 14 days post infection. Infections were scored as heavy, intermediate, weak, or uninfected as described[60]. *P*-value is shown for Fisher's exact test, two-sided. **b** Prevalence and intensity of proventriculus infections. For flies with a heavy midgut infection on day 14, the proventriculus was separated from the midgut and intensity of infections was scored using the same criteria as for the midguts. *P*-value is shown for Fisher's exact test, two-sided

simply due to overexpression of PDEB1, because the same construct did not influence SoMo when transfected into WT cells (Supplementary Figure 4).

Flies infected with either WT, KO, AB1, or AB2 cells were assessed for midgut infection at days 3, 7, and 14, and for proventriculus infection at day 14. In this case, differences in midgut infection rates and intensities (Fig. 5a) were less pronounced than observed in the initial fly infection experiment and were not statistically significant (Fisher's exact test, two-sided; Fig. 5a). For each fly with a heavy midgut infection on day 14, the proventriculus was also examined. Most of the flies infected with WT also had a heavily colonized proventriculus, while the KO mutant showed a complete failure to infect the proventriculus (p = 0.0001, Fisher's exact test, two-sided). Both AB1 and AB2 infected the proventriculus at rates comparable with WT (Fig. 5b). Therefore, the defect in colonization of the proventriculus can be attributed specifically to loss of PDEB1.

One possible explanation for the inability of KO cells to reach the proventriculus is that they do not differentiate from early to late procyclic forms in the fly midgut. To assess this, we examined expression of the early procyclic marker GPEET, which is expressed for the first 4–7 days post infection and is then switched off[7]. GPEET expression was monitored by IFA over the course of fly infection (Fig. 5c, d). The WT, KO, AB1, and AB2

were all ≥ 95% GPEET-positive at day 3, indicating that they were early procyclic forms, as expected. For all lines, GPEET was downregulated with similar kinetics. There was some variability at day 7, but this did not correlate with the presence or absence of PDEB1. Thus, the failure to populate the proventriculus is unlikely to be due to a defect in differentiation to the late procyclic form.

**The PDEB1 knockout is not complemented by wild-type cells.** Prior work with fluorescently labeled cells showed trans-complementation of the SoMo defect in the PDEB1 RNAi line by WT cells, because a mixed population was able to form radial projections with RNAi and WT cells migrating together in SoMo assays[34]. To determine whether WT cells could rescue the SoMo defect of KO cells, we used the WT-dsRed or KO-dsRed cells described above, mixed with unlabeled WT cells. Prior to performing these experiments, we mixed WT-dsRed or KO-dsRed with untagged WT cells and monitored their growth in suspension culture over a period of 14 days. Pure cultures of WT-dsRed and KO-dsRed were included as controls to assess spontaneous loss of the fluorescent protein. dsRed expression was very stable, with ≥ 94% of the population remaining fluorescent over 14 days (Fig. 6a). As predicted from the slightly longer population doubling times of tagged cells (Fig. 2b), these were progressively overgrown by untagged cells in mixed cultures. After 14 days, cultures initiated at a ratio of 2:1 (tagged:untagged) had 23% (WT-dsRed) and 20% (KO-dsRed) fluorescent cells, respectively (Fig. 6a). This 2:1 ratio was used for all subsequent mixing experiments.

KO-dsRed or WT-dsRed were co-cultured on plates with untagged WT cells. Both mixtures engaged in SoMo and formed projections (Fig. 6b). When imaged by fluorescence microscopy, WT-dsRed cells were evenly distributed throughout the projections (Fig. 6b, upper panel). However, in contrast to the PDEB1 RNAi line, a steep gradient of KO-dsRed cells was observed. Although KO-dsRed cells did move into the projections and a few could be seen at the tip, there was a sharp fall-off as the projection extended away from the center (Fig. 6b, lower panel). These results imply that cells that completely lack PDEB1 have an impaired response to external signals, in contrast to the RNAi line that still expresses PDEB1 at 10% of the level of WT cells[34].

It is not well understood how trypanosomes move from the midgut to the proventriculus. Given that interactions with surfaces in vitro[35] and with tsetse tissues in vivo[12] promote trypanosome collective behavior, the parasites could migrate individually or cooperatively. Alternatively, some leader cells might pave the way for other cells to follow as seen for example, during lateral line development in zebrafish[42]. We therefore asked if WT or PDEB1 KO cells impact each other's ability to infect the tsetse fly proventriculus. To assess this, coinfection experiments were performed in which flies were infected either with WT-dsRed alone, KO-dsRed alone or mixtures of WT-dsRed + untagged WT (WT + WT-dsRed), or KO-dsRed + untagged WT (WT + KO-dsRed).

On day 14, midgut infection rates with WT-dsRed alone (45%) and KO-dsRed alone (25%) (Fig. 6c) were consistent with our earlier observations using untagged parasites (Figs. 3 and 5). Likewise, in flies with midgut infections, proventriculus infection rates were similar to those obtained with untagged cells, and there was a clear difference between WT-dsRed (83%) and KO-dsRed (18%). The intensity of proventricular infection was also reduced in the KO relative to WT, as observed for untagged cells. Therefore, dsRed cells are reliable reporters of proventriculus infection. The primary objective of these experiments was to determine whether WT cells could facilitate proventriculus

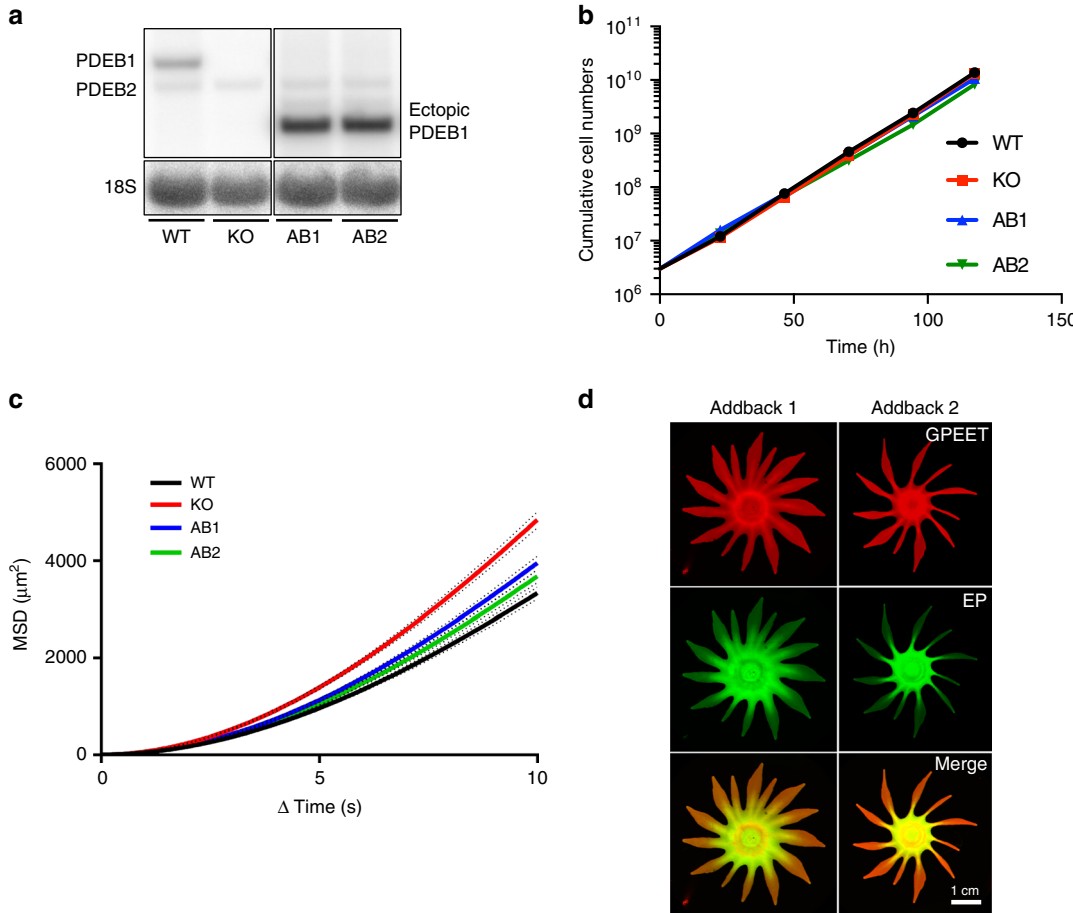

**Fig. 4** Ectopic expression of PDEB1 in the PDEB1 KO restores SoMo. **a** Northern blot analysis of RNA from WT, PDEB1 KO and two addback clones, AB1 and AB2, expressing an ectopic copy of PDEB1 that is integrated upstream of a procyclin locus and transcribed from the procyclin promoter[44]. The addback version has a truncated 3'-untranslated region derived from the EP1 procyclin gene, hence the smaller size of the transcript compared with the endogenous copy. The probe used is described in Fig. 2. 18S rRNA serves as a loading control[66]. The expression of the ectopic copy relative to the endogenous copy is 4.3-fold for AB1 and 4.2-fold for AB2. **b** Growth of WT, PDEB1 KO, AB1, and AB2 parasites was monitored over the course of 5 days in suspension culture as described in Fig. 2b. **c** MSD is measured for WT, PDEB1 KO, AB1, and AB2 cells from two biological replicates. Results are from $n = 1919$ tracks for WT, $n = 1824$ tracks for PDEB1 KO, $n = 1876$ tracks for AB1, and $n = 1973$ tracks for AB2 from a total of 34 videos for each. Dotted lines represent standard error of the mean (SEM). **d** Community lifts demonstrate addbacks AB1 (left) and AB2 (right) are SoMo-positive and express both GPEET (red) and EP (green)

infection by the KO. We therefore examined proventriculi of flies infected with KO-dsRed parasites alone versus flies coinfected with WT + KO-dsRed. Since we have never observed a proventricular infection in the absence of a midgut infection, we limited our analysis to flies that were midgut-positive (Fig. 6d and Supplementary Figure 5). Coinfection did not influence the proventricular infection rate by KO-dsRed cells, as the KO-dsRed infection rate in mixed infections (18%) was the same as that observed in flies infected with KO-dsRed alone (Fig. 6d). Importantly, nine of these seventeen proventriculi were also assessed for total parasites, and all nine showed heavy infection by untagged WT parasites (Supplementary Movie 4). As a control to determine whether dsRed cells might simply be outcompeted by untagged cells in the fly, we assessed the prevalence of WT-dsRed parasites in proventriculi of flies coinfected with WT-dsRed + untagged WT. The prevalence of WT-dsRed in the proventriculus for mixed infections was 85%, nearly identical to that for WT-dsRed cells alone (83%), indicating that competition was not an issue (Fig. 6d).

When individual proventriculi were examined, heavy infection by WT-dsRed cells was clearly evident, and parasites were distributed throughout (Fig. 7a), although the thickness of the

sample makes this difficult to see in a single focal plane. Z-stacks are shown in Supplementary Movies 5 and 6. On the other hand, KO-dsRed cells, if detected at all, were present in extremely low numbers, corroborating earlier experiments with untagged parasites and directly illustrating that reduced prevalence of proventriculus infection is accompanied by reduced intensity of infection. The coinfection experiments with KO-dsRed and WT parasites enabled us to analyze the behavior of the two genotypes within single flies. These experiments demonstrated that, despite having only one or two KO parasites, proventriculi were heavily infected with WT parasites (Supplementary Movie 4). These results prove first, that the poor infection rate with the KO is not because the fly is refractory to proventricular infection per se, and second, that WT and KO parasites act independently of each other. Individual KO-dsRed parasites are motile within fly tissue (Supplementary Movie 4), supporting earlier in vitro experiments that showed PDEB1 is not required for motility of individual cells, despite being necessary for collective motility of the group. Taken together, these data are consistent with the idea that PDEB1 KO is unable to respond to external signals; these signals might emanate from the fly and/or other parasites.

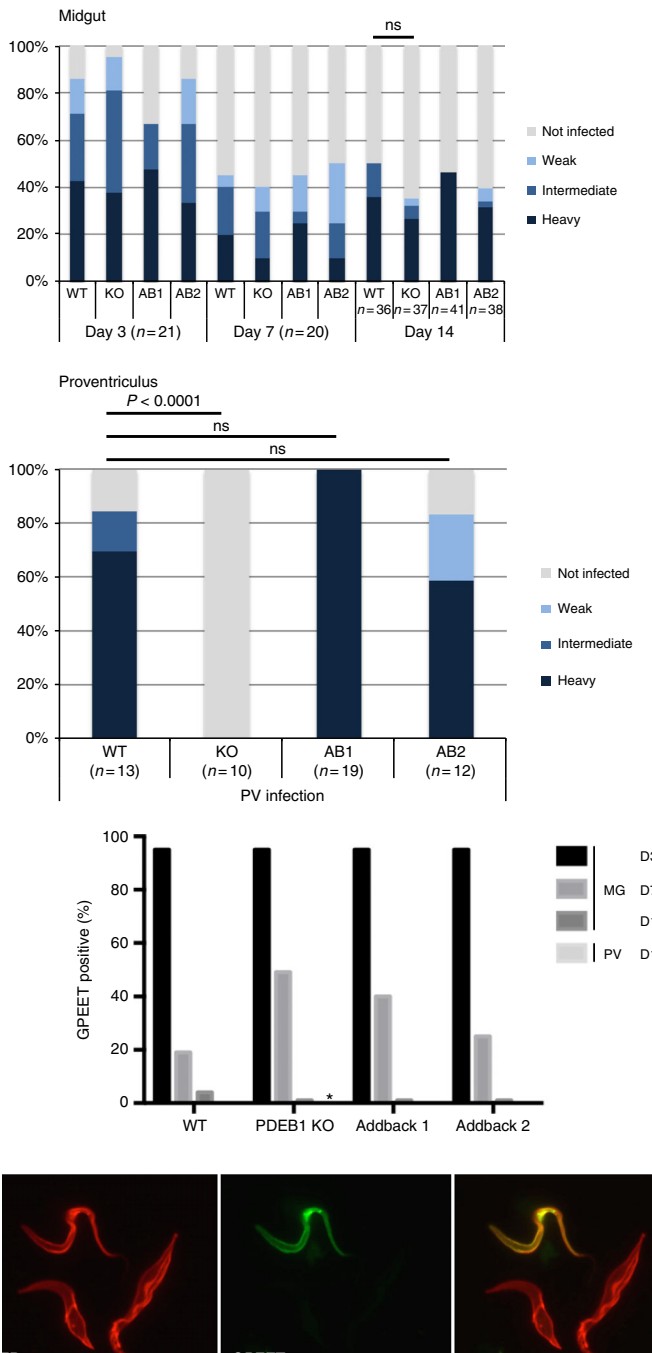

**Fig. 5** Addback of PDEB1 restores invasion of the proventriculus. **a** Teneral flies were infected and the prevalence and intensity of infection were determined as described in Fig. 3a. Day 14 infection rate was not significantly different between WT and KO (ns, Fisher's exact test, two-sided). **b** Prevalence and intensity of proventriculus infections in flies with a heavy midgut infection on day 14. Tissue samples were processed as described in Fig. 3b. P-value is shown for Fisher's exact test, two-sided, ns: not statistically significant. **c** Kinetics of GPEET repression in vivo. Parasites from the midgut (days 3, 7, and 14 post infection) and the proventriculus (day 14 only) were stained with anti-GPEET and anti-EP antibodies. One hundred cells were counted per sample. The percentage of GPEET-positive trypanosomes is shown. All parasites were EP-positive. *: no cells were detected in the proventriculus of the KO. **d** Representative immunofluorescence image of WT parasites taken from the midgut at day 7 post infection and probed with the indicated antibodies. Scale bar: 10 microns

**PDEB1 is needed for transition to the ectoperitrophic space.** The finding that the PDEB1 KO mutant had a pronounced defect in establishing a proventricular infection, even in flies with heavy midgut infections, indicated that either the transition from the midgut lumen to the ectoperitrophic space or from the ectoperitrophic space to the proventriculus (Fig. 1) is dependent upon cAMP signaling. In standard dissections, the midgut is considered as a single unit and there is no information on whether parasites are in the lumen or ectoperitrophic space. To distinguish between these compartments, infected flies were fed fluorescently labeled wheat germ agglutinin (FITC-WGA) 24 h before dissection. FITC-WGA binds the PM, delineating the border between the midgut lumen and ectoperitrophic space (Fig. 1b); when combined with infection by fluorescent parasites, this allows us to investigate their topological distribution within the midgut. At 14–15 days post infection, fly midguts and proventriculi were carefully removed, so that both tissues remained connected and unbroken, and embedded in low melting agarose. Tissue samples were also labeled with Hoechst to visualize nuclei of tsetse fly epithelial cells and examined by fluorescence microscopy. In WT-dsRed-infected flies, parasites could clearly be seen throughout the midgut, including in the ectoperitrophic space (Fig. 7b; Supplementary Movies 7, 8, and 9). These results confirm that infection of the ectoperitrophic space is well-established by day 14. In contrast, in KO-dsRed-infected flies, parasites were rarely detected in the ectoperitrophic space, even when the midgut was heavily infected (Fig. 7b). Of 24 infected flies, 13 had no parasites in the ectoperitrophic space, and in the other 11 flies they were few in number. From these results, we conclude that PDEB1-dependent cAMP signaling is required for parasites to successfully transition from the midgut lumen to the ectoperitrophic space (see step 1 of Fig. 1a).

## Discussion

*T. brucei* transmission through the tsetse requires parasite movement through diverse tissues and is accompanied by an ordered series of parasite developmental changes in specific tissues. It is therefore hypothesized that the parasite employs specific signal transduction pathways to sense and respond to different extracellular environments encountered in these tissues. Our results provide the first formal evidence in support of this hypothesis and, additionally, link the requirement for cAMP signaling in SoMo to a specific step in the parasite life cycle.

The primary defect of PDEB1 KO was its inability to make the transition from the midgut lumen to the ectoperitrophic space. To our knowledge, this is the first mutant demonstrated to be defective at this step of the transmission cycle. It has been reported previously that the transition from the midgut to the proventriculus is not a bottleneck for WT trypanosomes[13,14]. Our results are consistent with this, as > 80% of WT-infected flies with a midgut infection also had a proventriculus infection. Our findings indicate, however, that the PM presents a formidable barrier that must be actively overcome, because loss of PDEB1 prevents parasites from establishing infection in the ectoperitrophic space. The infection defect was not due to a block in parasite differentiation or motility, because the KO showed normal kinetics of differentiation from early to late procyclic forms within the fly and was fully capable of processive motility in liquid culture. Rather, our results indicate the defect results from disruption of cAMP signaling necessary for progression from the midgut lumen to the ectoperitrophic space. At present, our data do not distinguish between a defect in physical crossing of the PM, versus a defect in survival once the ectoperitrophic space is reached. We favor a defect in crossing, first because the few parasites that do get across are alive and motile (Supplementary

 7

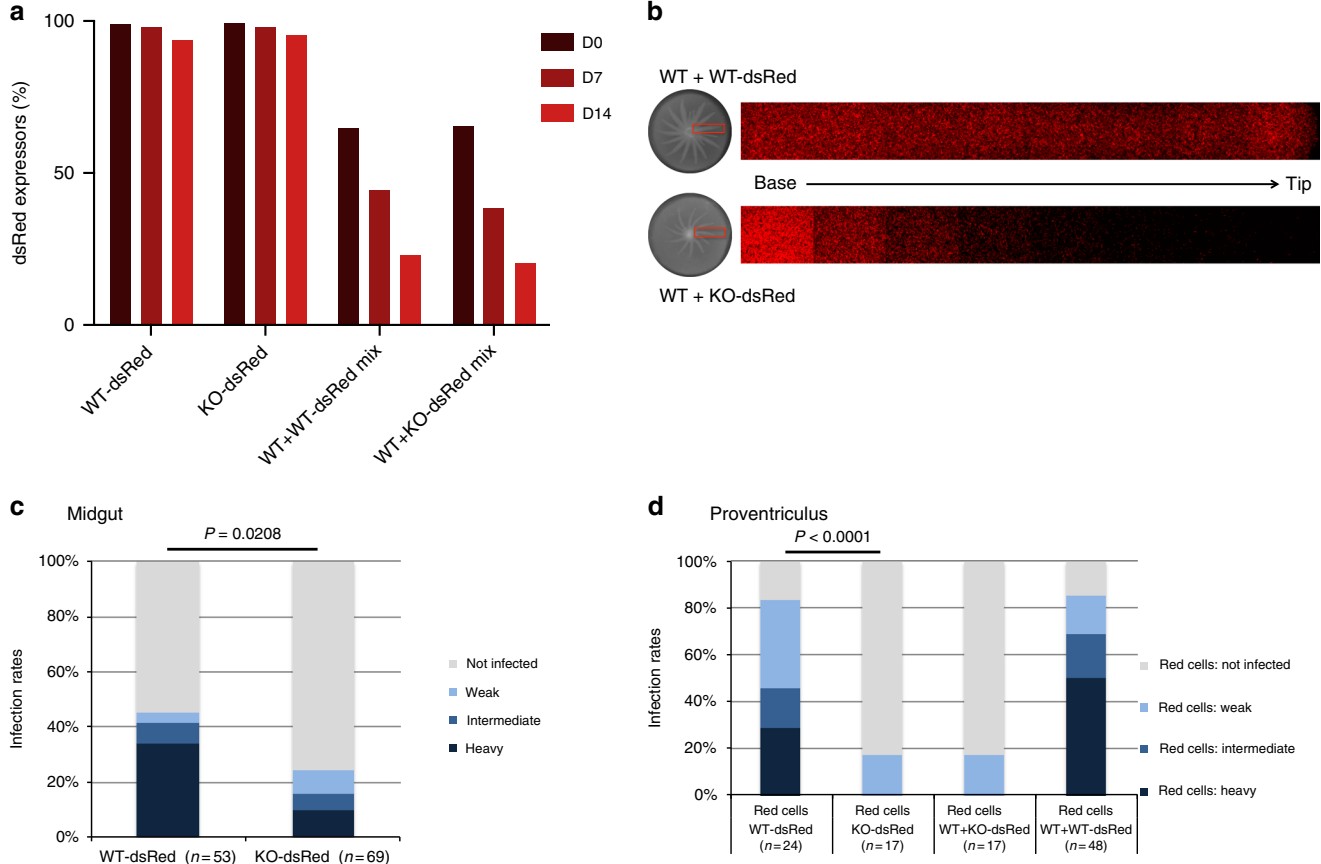

**Fig. 6** WT does not complement PDEB1 KO in vitro or in vivo. **a** Proportion of dsRed-positive cells in suspension culture. Fluorescence was monitored by flow cytometry (Supplementary Figure 2). Tagged:untagged cells were mixed at a ratio of 2:1. **b** WT-dsRed or KO-dsRed was mixed with untagged WT at a ratio of 2:1 and tested for SoMo. Top panel: WT + WT-dsRed co-culture. WT-dsRed parasites are evenly distributed throughout the projections. Lower panel: KO + WT-dsRed co-culture. KO-dsRed cells form a gradient with a high proportion of KO-dsRed cells at the base of the projection and progressively fewer cells extending to the tip. **c** Prevalence and intensities of midgut infections at day 14 post infection. P-value is shown for Fisher's exact test, two-sided. **d** Proventriculus infection rates at day 14 post infection. In the mixed infections, only dsRed cells were scored. P-value is shown for Fisher's exact test, two-sided

Movie 4), second, the inability to move between compartments correlates with altered movement in vitro, and third, there is a precedent for cyclic nucleotides controlling cell movement in a broad range of organisms[16–18].

It is premature to say whether the coordinated movement exhibited by trypanosomes during SoMo takes the same form as in the insect vector. Recent work has demonstrated trypanosomes exhibit coordinated flagellar beating within tsetse tissues, but the shorter timescale indicates a different mechanism than used for SoMo[12] and it is unknown whether these two types of collective behavior are connected. Nonetheless, our findings demonstrate that SoMo in vitro depends on activities that are also required in vivo. These entail movement and the ability to respond to signals that might alter the form or direction of motility, i.e., chemotaxis. The reduced turning frequency of PDEB1 KO compared with WT is similar to what is seen with bacterial chemotaxis mutants[39]. This would support that the primary defect lies in responding to signals that control motility. This response could be individual or collective.

Simultaneous visualization of host tissues and individual trypanosomes allowed us to analyze parasite distribution in the lumen and ectoperitrophic space. In addition to defining the step at which the PDEB1 mutant was blocked, this has delivered new insights into how infections develop. When teneral flies are challenged with trypanosomes, approximately half of them manage to eradicate the infection after 6 days[41]. It has been tacitly assumed that trypanosomes had to colonize the ectoperitrophic space in order to establish a chronic infection, and it was unknown whether trypanosomes remained in the lumen for the duration of the infection[43]. By using a fluorescent lectin to label the PM in combination with fluorescent parasites, we showed that the PDEB1 KO could proliferate and persist in the lumen without colonizing the ectoperitrophic space. Furthermore, numerous WT trypanosomes were present in the lumen as well as in the ectoperitrophic space 14 days post infection. At this time point, they were all late procyclic forms. These findings challenge the paradigm that trypanosomes must gain access to the ectoperitrophic space to maintain a midgut infection, and underline the importance of discriminating between the different sub-compartments in the midgut. It is worth noting that a peritrophic matrix is a common feature among blood-feeding insects, including mosquitos that transmit malaria. Therefore, our findings have relevance for understanding transmission biology of other vector-borne diseases.

Several earlier studies have employed deletion mutants to study trypanosome–tsetse interactions, but these have been restricted to the midgut (as a whole) and the salivary glands[32,44–46], without examining the intervening tissues. In the light of our finding that

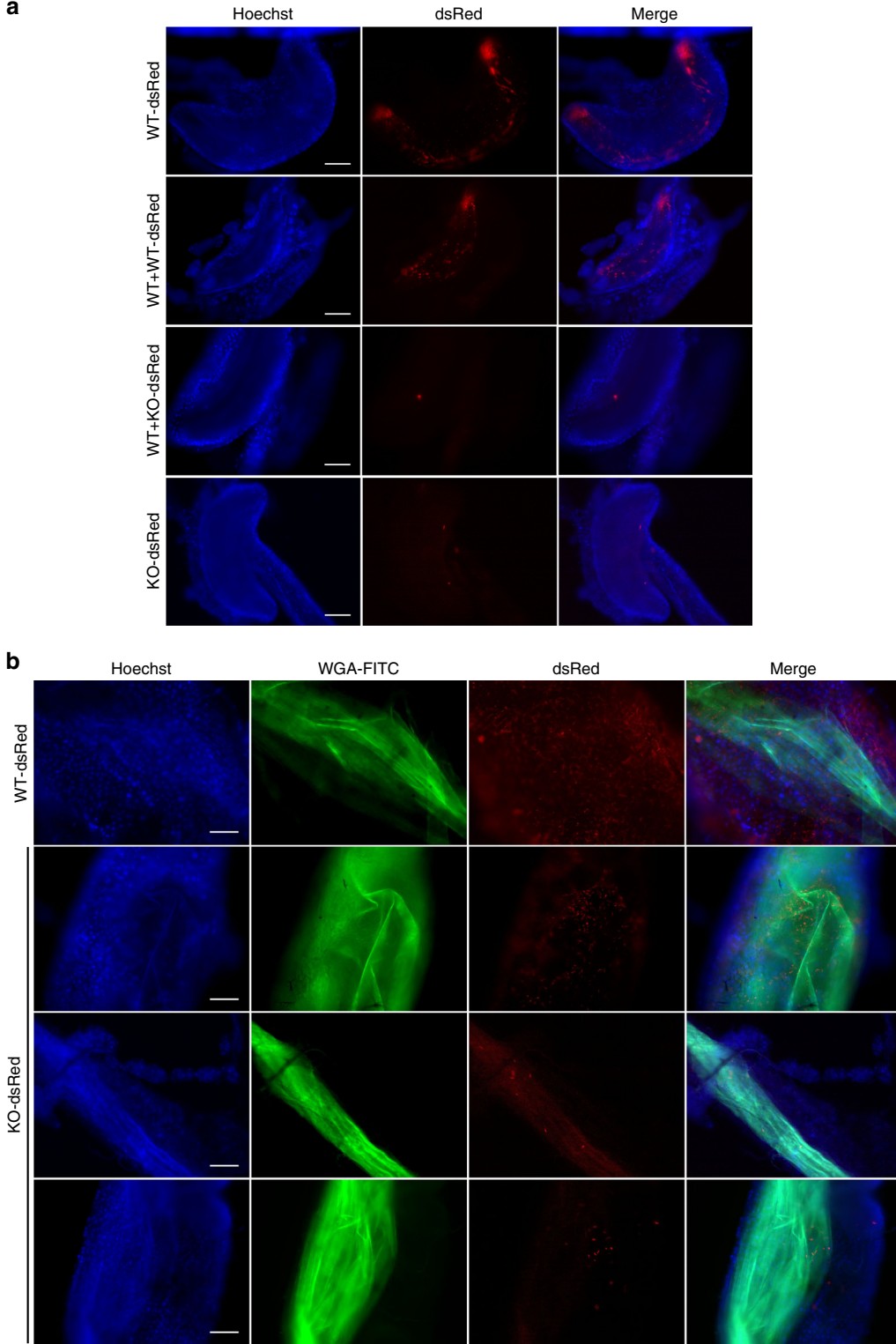

**Fig. 7** Impaired cAMP signaling impacts the prevalence and topology of infection. **a** Proventriculi at day 14 post infection. Nuclei of host cells are stained with Hoechst dye. In mixed infections (middle panels), only the dsRed-tagged cells are visible. Supplementary Movie 4 shows merged DIC and red fluorescent channels for a mixed infection with WT + KO-dsRed. **b** Midguts of flies infected with dsRed-tagged trypanosomes and fed with FITC-WGA 24 h prior to dissection at day 14. Nuclei of host cells are stained with Hoechst dye. Images in **a** and **b** were captured using a TillPhotonics/FEI iMIC digital spinning disc microscope. Scale bars: 100 microns

the transition from the midgut lumen to the ectoperitrophic space requires specific parasite factors, it will be worth re-examining some of those mutants. For example, trypanosomes lacking PDEA were still able to infect the midgut[32], but it is not known whether they are able to exit the lumen. Another mutant, Δproc, which lacks all procyclin genes[46,47], showed a mild defect in establishing midgut infections, but was 10 times worse than WT at colonizing the salivary glands[46]. This result seemed

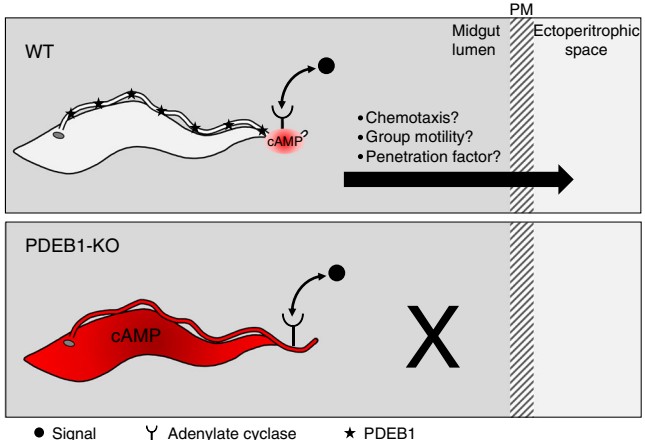

**Fig. 8** Model for infection defect of PDEB1 knockout parasites. In wild-type *T. brucei* (WT), signals in the midgut lumen (black circle) modulate cAMP production (red) by receptor-adenylate cyclases located in specific regions of the flagellum. PDEB1 is distributed along the flagellum (black stars) and restricts cAMP to site of production by the AC, where local changes in cAMP concentration control chemotaxis, group motility or other factors that facilitate traversal of the peritrophic matrix (PM). In the absence of PDEB1 (PDEB1-KO), cAMP levels rise and diffuse through the flagellum and cell, so the parasite is no longer able to generate localized cAMP fluctuations and is thus unable to respond to signals that direct traversal of the PM

paradoxical, given that procyclins are not expressed by salivary gland trypanosomes. EP procyclin is expressed by parasites in the ectoperitrophic space and proventriculus, however, so it is possible that the null mutant has difficulty leaving the lumen or gaining access to the proventriculus.

Our findings suggest that there might be a limited window of opportunity when early procyclic forms are able to egress from the midgut lumen to the ectoperitrophic space. Once the parasites differentiate to late procyclic forms, they can survive in the lumen, but they may not be able to move on and complete the cycle. Successful crossing of the PM could be very rapid, as there are no publications that show trypanosomes in the process of entering the ectoperitrophic space. There are two reports showing electron micrographs of trypanosomes between the layers of the PM[15,48], sometimes in cyst-like structures. In the latter case, these were seen 40 days post infection, and it is open to debate whether they are intermediates in the process of crossing the PM or dead ends. Continued technological advances in imaging, together with the appropriate choice of time point after infection, may allow these events to be analyzed and to determine whether the parasites make this transition individually or as a group.

Our results provide the first example of tissue-specific requirements for the cAMP signal transduction pathway in the trypanosome life cycle. cAMP is produced by receptor-type ACs that are at the surface membrane. Different AC isoforms are expressed in different life-cycle stages[24,27] and by trypanosomes in different tsetse tissues[26]. Thus, while signaling molecules remain to be identified, receptor ACs are well-suited to transduce tissue-specific responses. Trypanosome cAMP signaling is thought to act through downstream effectors, termed cAMP response proteins (CARPs)[49]. At least two CARPs, together with PDEB1 and all ACs examined so far, are restricted to the flagellum[23,24,49,50]. Our findings therefore support the paradigm of the flagellum (also known as the cilium) as a signaling platform for directing cellular adaptation to changing extracellular

conditions[51] and extend this paradigm to a group of devastating pathogens.

A mechanistic explanation for the infection defect of PDEB1 mutants is provided by considering PDEB1 function in cAMP signaling (Fig. 8). Flagellar PDEB1 is postulated to provide a barrier to diffusion of cAMP that restricts signal transduction to the site of cAMP generation[34,50]. Pharmacological inhibition of PDE increases intracellular cAMP[34,50]. One could therefore imagine that loss of PDEB1 floods the flagellar compartment with cAMP, disrupting highly localized and insulated signals originating from specific ACs that are distributed to specific regions of the flagellum[24]. This would prevent the parasite from properly interpreting signals received in the midgut lumen that enable it to cross the PM and continue the transmission cycle. *T. brucei* encodes two different PDEB isoforms[22]. Prior work has shown that PDEB2 compensates for PDEB1 in maintaining parasite viability in bloodstream forms[50], suggesting some overlap in function for these two proteins. However, our results show that PDEB2 is unable to compensate in the context of tsetse fly infection, thereby demonstrating isoform-specific functions.

The transition of the parasite from one tissue to the next has been described evocatively as a series of gates under the control of the fly[13]. Based on our results, it would seem that while the fly might be the gatekeeper, the parasite employs cAMP signaling to engage keys that unlock these gates. This might also apply in the mammalian host, where the parasite can breach the blood–brain barrier or take up residence in the adipose tissue or skin[52–55]. Given that host and parasite PDEs can be differentially inhibited[56,57], our findings might have wider implications for development of therapeutics.

## Methods

**Trypanosomes.** *T. brucei brucei* Lister 427 and derivatives thereof were used in this study. Procyclic forms were cultured in SDM79[58] or SDM80[59] plus/minus glucose (5.55 mM) containing 10% heat-inactivated fetal bovine serum (FBS) at 27 °C and 2.5% $CO_2$. Procyclic forms of the Bernese stock of Lister 427 can be maintained as early procyclic forms in the absence of glycerol[40,60]. Parasites were maintained at a density between $10^6$ and $10^7$ cells ml$^{-1}$. Population doubling times were determined over a period of at least 5 days in which the cell density was determined daily.

**Plasmids and generation of knockout and addback clones.** The PDEB1 KO was generated by two rounds of homologous recombination using genes conferring resistance to blasticidin and puromycin, respectively. The resistance genes, flanked by 452 bp upstream and 635 bp downstream of the PDEB1 coding region, were cloned between the Xho I and Bam HI restriction sites in the pTub plasmid backbone[61,62], thus removing the tubulin sequences. The following primers were used to amplify the PDEB1 flanking sequences:
Upstream FWD: atatGCGGCCGCTGCATTATGTTACTTGGGGGCA
Upstream REV: atatCTCGAGGACGTAGTGTCCAACTGTGC
Downstream FWD: atatGGATCCAGTCAGTTGACCGGTGGTAG
Downstream REV: atatTCTAGACCGCCACAACTCCCTCTTAC
Plasmids were digested with NotI-HF and XbaI to release the insert prior to transformation. The knockout was verified by PCR using primers:
P1 (PDEB1_ORF_fwd): AGTACTCATTGTGCACAGTT
P2 (PDEB1_ORF_rev): TCATGTATATCTGTAGGCAT
P3 (PDEB1_KO_fwd): TCATACGGCTATTTGCCCAGT
P4 (PDEB1_KO_rev): AATGTCACACAACCGCAGTG
A PDEB1 addback plasmid was generated by inserting the coding region between the EcoRI and BglII sites in pGAPRONE-mcs[63]. The coding region was amplified with the following primers:
Forward: atatGAATTCATGTTCATGAACAAGCCCTTTG
Reverse: atatAGATCTTCAACGAGTACTGCTGTTGTTG
The construct was linearized with SpeI, enabling integration upstream of a procyclin locus[44].
Transfection was performed as described previously[60]. Stable transformants were selected using 10 μg ml$^{-1}$ blasticidin, 1 μg ml$^{-1}$ puromycin or 15 μg ml$^{-1}$ G418.
Construction of pTB011_Cas9_T7RNAP_blast: the plasmid pTB011[64] was digested with Nco I to excise the puromycin *N*-acetyl-transferase (pac) gene and 350 bp of the upstream alpha-tubulin flanking region. In a second step, the alpha-tubulin flanking region and the open-reading frame (ORF) of T7RNA polymerase

were amplified separately by PCR. The vector backbone, the tubulin fragment and the ORF of T7RNA polymerase were cloned together by Gibson assembly. The plasmid was digested with Pac I to release the insert prior to transformation. Transfection was performed as described previously[60]. Stable transformants were selected using 10 µg ml$^{-1}$ blasticidin.

Primers used to generate individual fragments for the Gibson assembly:
Cas_P1 GibAs: acgtcgcatgctcccggccgccatggccgcgggattttaa
Cas_P2: gcgatgttaatcgtgttcatgaattcgtttgaactatttt
Cas_P3: aaaatagttcaaacgaattcatgaacacgattaacatcgc
Cas_P4 GibAs: caactaaatgggcaccatggttacgcgaacgcgaagtccg

Trypanin knockout: PCR amplification of targeting fragments and sgRNA templates were performed as described previously[64]. To delete trypanin, targeting cassettes were amplified from pPOTv7-hygromycin and pPOTv7-G418[64]. The knockout was verified by PCR using primers: Tryp-5Flk_Fw: GCTGAGATAGTTTAAGAGGGAGAG and Tryp-ORF_Rv: GACATATGCTACTCAAAGTTGCTCCGTG.

Primers used for PCR amplification of targeting fragments:
Trypan-crKO_Fwd:
TACTTTTCAGACTGCATCGTGGCGTACCCCgtataatgcagacctgctgc
Trypan-crKO_Rev:
CTGCAACAAAGCCGTAACTTGGAACAACCAccggaaccactaccagaacc

Primers used for PCR amplification of sgRNA templates:
Trypan-crKO_5gR:
gaaattaatacgactcactataggCAAAAACGAGAAGAGCCTACGgttttagagctagaaatagc
Trypan-crKO_3gR:
gaaattaatacgactcactataggAGGTGTTGTGGTTCACACGTgttttagagctagaaatagc

**RNA isolation and northern blot analysis**. Total RNA isolation and northern blot analysis were performed according to standard procedures[65]. Ten micrograms of total RNA were loaded per lane. Radioactively labeled probes were generated using a Megaprime DNA-labeling system (Amersham Biosciences, Buckinghamshire, UK) according to the manufacturer's instructions. Blots were hybridized and washed under stringent conditions. 18S rRNA, detected with a 5′-labeled antisense oligonucleotide, was used as a loading control[66]. Signals were normalized in Fiji (Version 1.0)[67].

**Flow cytometry**. Flow cytometry (NovoCyte, ACEA Biosciences, Inc., San Diego, USA) was used to monitor the proportion of GPEET-positive or dsRed-positive cells. GPEET was detected with rabbit polyclonal anti-GPEET as described[40].

**Social motility assay**. SoMo assays were performed as described[36], but without the addition of glycerol. In complementation experiments, either WT-dsRed or KO-dsRed were mixed with WT cells at a ratio of 2:1. Two hundred thousand cells were used as the inoculum. Imaging of fluorescent cells was performed using a Leica DM 5500 B microscope at x20 magnification. Community lifts for the detection of GPEET and EP procyclins were performed as described[36].

**Community lifts**. Community lifts for the detection of GPEET and EP procyclins were performed as described[36] using K1 rabbit anti-GPEET at a dilution of 1:1000 and TBRP1/247 mouse anti-EP (Cat. no. CLP001A, Cedarlane Laboratories, Burlington, Canada) at a dilution of 1:2500 as primary antibodies. The secondary antibodies goat anti-mouse IRDye 800CW (LI-COR Biosciences, Bad Homburg, Germany) and goat anti-rabbit IRDye 680LT (LI-COR Biosciences) were used at dilutions of 1:10,000.

**Fly infection and staining of the peritrophic matrix**. *Glossina moristans moristans* pupae were obtained from the Department of Entomology, Slovak Academy of Science, Bratislava, Slovakia. Teneral flies were infected by membrane feeding with $2.5 \times 10^6$ parasites ml$^{-1}$ and maintained as described[60]. For complementation experiments, dsRed-tagged and untagged cells were mixed at a ratio of 2:1. The intensities of midgut infections were graded as described[60]. The infective feed was performed with washed horse red blood cells resuspended in SDM79; all subsequent feeds consisted of whole defibrinated blood (TCS Biologicals, Buckingham, UK).

Staining of the peritrophic matrix: At days 10–13 post infection, flies were collected and fed 40 µg ml$^{-1}$ fluorescein wheat germ agglutinin (WGA-FITC; Adipogen, Liestal, Switzerland) diluted in SDM79 supplemented with 5% defibrinated horse blood and 10% FBS. Twenty-four hours later, intact fly midguts, still connected to the proventriculus, were removed, placed on a coverslip, submerged in 20 µg ml$^{-1}$ Hoechst dye in PBS for 1 min, and embedded in 1% low melting agarose. Images and videos were captured using a TillPhotonics/FEI iMIC digital spinning disc microscope.

**Motility traces**. Motility assays were performed in motility chambers[68] using a Zeiss Axiovert 200 M inverted microscope at x10 magnification. WT, PDEB1 KO, or TPN KO cells in suspension culture were imaged at 30 frames per second using Adobe Premiere Elements 9. A total of 34 videos each for WT, PDEB1 KO, and TPN KO were analyzed from three biological replicates. In a separate experiment,

34 additional videos each for WT, PDEB1 KO, Addback 1, and Addback 2 were analyzed from two biological replicates. Mean-squared displacement of individual cells was determined using a trypanosome-specific cell tracking algorithm developed in MATLAB[69] based on a single-particle-tracking algorithm[70]. We used a maximum time interval of 10 s and only considered cells that were in focus for a minimum of 300 out of 900 frames.

**High-speed cell imaging**. Cells were imaged in motility chambers as described above with the modification of placing cells between two coverslips separated by double-stick tape instead of a microscope slide and a coverslip separated by double-stick tape. Videos were taken on an Olympus IX83 microscope using phase contrast with a x10 objective lens and x2 magnification. Videos were taken on a Hamamatsu Orca Flash 4.0 camera at 496 frames per second. Videos were captured as image stacks in MetaMorph Advanced. Image stacks were converted to AVI videos using Fiji-ImageJ Version 1.0[67]. Still images in Fig. 2d are taken from the original image stacks. Four to five videos each were taken of WT, PDEB1 KO, and TPN KO cells and these videos were concatenated in a single Supplementary Movie for each cell line using Adobe Premier Elements 14 (WT = Supplementary Movie 1; PDEB1-KO = Supplementary Movie 2; TPN-KO = Supplementary Movie 3).

**Reporting summary**. Further information on experimental design is available in the Nature Research Reporting Summary linked to this article.

## Data availability

All data are available from the authors. The relevant source data underlying all relevant figures are provided as a Source Data file. Figures with raw data are: Fig. 2a–c, Fig. 3a, b, Fig. 4a–c, Fig. 5a–c, Fig. 6c, d, Suppl Fig. 1, Suppl Fig. 2h–i, Suppl Fig. 3a–b, Suppl Fig. 4a, Suppl Fig. 5.

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

## Acknowledgements

Michelle Shimogawa, Simon Imhof, and Gaby Schumann are thanked for their insightful comments on the paper. Markus Engstler and Sara Schuster are thanked for helpful

discussions and providing protocols for PM staining. Nicholas Doiron is thanked for assistance in dissecting flies. Aydogan Ozcan, Hatice Koydemir, and Muhammed Veli are thanked for their assistance with high-speed imaging and Sebastian Knüsel and students of the Bachelor Practical 2018 for help with generating the trypanin knockout. K.H.: NIH grant AI052348. I.R.: Swiss National Science Foundation grant 31003A_166427 and HHMI grant 55007650. S.D.: Ruth L. Kirschstein National Research Service Award GM007185 and Ruth L. Kirschstein National Research Service Award AI007323.

## Author contributions

S.S., S.D., I.R., and K.H. designed the experiments and wrote the paper. S.S., S.D., R.R., T.W. and F.F. conducted the experiments. S.S., S.D., T.W., I.R. and K.H. analyzed the data. K.H. conducted the statistical analyses.

## Additional information

**Competing interests:** The authors declare no competing interests.

