## [Peer Review File · Nature Communications]

Reviewers' comments:

Reviewer #1 (Remarks to the Author):

This is a nice attempt to relate some in vitro observations to the biology of the Tsetse lifecycle of the trypanosome parasite. It's important that people are starting to generate mutants and seeing how they behave in in vivo environments to try and tease out phenotypes not seen in culture. The Given that direct observations of the biochemistry in the infected fly is so difficult the conclusions depend upon a set of correlations between different in vitro and in vivo characteristics and measurements (motility and biochemistry in vitro and infectivity of gut areas in the in vivo). These correlations need to be very firmly based to rule out other possibilities in order that causal conclusions can be drawn. This is nice work but needs a little more to rule out some other possibilities – namely a cryptic motility defect or a differentiation (early / late procyclic) defect rather than a social motility cause.

Therefore I do have some reservations-

1. The title needs changing – the work here is not about navigation through tissues.
2. Given the work on genetics of motility phenotype defects in other systems such as Chlamydomonas a rigorous assessment of motility is critical for the statement to be made that this is about flagellar signalling and not a motility defect. This is too important a result that the only data presented on the mutant movement is the graph of mean square displacement. They have the videos so I one would also expect to see the traces of the cell movement analysed in terms of directional issues, processivity, etc. It is perfectly possible for the MSDS graphs to be similar but the mutant cells to exhibit other motility phenotypes. Also, the standard assessment of motility should also include an analysis of high speed movies of the cells to show that the flagella behave in the same way. The results here can still be explained by some subtle motility defect.
3. The lines 176 – 182 suggest that the worry is that the KO cells have become late procyclic forms since the Social Motility phenotype is restricted to early forms. The authors suggest this is not an issue since the KO express the GPEET marker. However, does this not raise the more fundamental issue of whether the mutant cells in the tsetse (albeit fewer) that are able to establish the late infections are indeed late procyclics. Ie that the mutants are defective in transition to the late forms in the fly and so there are less of them because of replication of this form not because of “navigation”? The type of parasites seen in the infections of both mutant and wild type surely need assessing because of this issue?
4. There is variability between the infection rates on different experiments (3a vs 5a), which is concerning with a significant infectivity effect seen in 3a but not 5a. Is there a general loss of fitness in the KO or not, the results make it difficult to be sure? Moreover, the use of t-tests seems inappropriate for analysing changes in categories a chi squared test would be better. The proventriculus effect does appear to be consistent between the experiments though.
5. The use of a northern blot to confirm the KO seems odd as this just shows lack of expression not loss of the gene - a Southern or PCR test is needed to show definitively that the gene is gone and not that its expression has stopped. In that phase (but possibly not others or not in vivo) of the life cycle.
6. I was confused by the timings in the discussion - early procyclic forms according to their data are lost by day 7 (fig. 5c - GPEET expression) yet on line 189 day 7 is when the trypanosomes cross the PM and enter the ectoperitrophic space - if this is the time that SOMO is required for movement across the PM why are they switching away from expression of GPEET to late procyclic markers and to a cell type not capable of SOMO?
7. Reading the literature about early and late parasites it appears that glycerol is required to maintain early PCFs yet here this stock of 427 cells are maintained as early PCFs without glycerol, which seems counterintuitive?
8. Given the authors earlier work and that of others on genetics of establishment of infections in the salivary gland how do the authors square the idea of mass movements via SoMo with the idea

that there are large bottlenecks and a founder effect.

9. It would've been very valuable to consider the tsetse infection phenotype of true (paralysed) motility defects relative to the SoMo defect they show. One issue is the general baseline for motility of any kind in this life cycle transition process. What is the magnitude of the defect relative to a cell that can't swim? This is essentially a question about the biological significance of the result.

Reviewer #2 (Remarks to the Author):

Trypanosoma brucei is transmitted by tsetse flies. Once inside the fly, the parasites have to overcome several challenges in order to survive and develop, including colonization of the midgut ectoperitrophic space. This step is essential for establishing an infection in the tsetse. How trypanosomes reach this compartment is still debatable, but the most accepted hypothesis is that they must cross the fly's peritrophic matrix (PM). In addition, it is well accepted that in vitro grown procyclic trypanosomes (the equivalent forms to the ones found in the tsetse gut) undergo social motility (SoMo). The molecular bases of trypanosomes SoMo are still unknown, but elegant work published by the same authors have shown that this phenomenon occurs only in early procyclic cells and that it may involve multiple pathways/genes, including cAMP signalling systems. In addition, whether SoMo is relevant for parasite colonization of the fly's gut has been a burning question within the trypanosome community. Thus, this is a relevant and important topic for understanding vector-parasite interactions in general.

In this MS the authors investigated the in vivo role of SoMo by infecting tsetse flies with a *T. brucei* phosphodiesterase PDEB1 deletion mutant, which shows a SoMo phenotype in vitro. By using live confocal microscopy analyses of labelled tsetse tissues and fluorescent parasites, the authors show evidence that PDEB1 mutants remained trapped in the tsetse gut lumen, suggesting a possible role in the migration to the ectoperitrophic space. However, although there is no doubt that this mutant shows an infection phenotype in the fly, the main problem of this paper is the bias interpretation that this is due to the lack of SoMo and not as direct consequence of having a defect in cAMP metabolism. Of course, defects in cAMP metabolism could lead to a SoMo phenotype, but so other other mutations that in principle are not metabolically related. Thus, given the lack of clarity and experimental evidence on the importance of trypanosome SoMo during colonization of the tsetse gut, the paper becomes merely descriptive and rather reports on the in vivo essentiality of another trypanosome gene.

Below, I list several other concerns and suggestions that need to be considered by the authors in order to show a stronger evidence of the potential role of trypanosome SoMo in the tsetse:

1) The environment of the tsetse midgut (including the type and availability of some nutrients) is very different to the nutritional environment of any procyclics culture media. This means that the authors may have overlooked a growth phenotype of the PDEB1 mutants if these parasites are cultured in chemically defined media. I wonder if at least the authors have tested the growth phenotype of this mutant in media with no or little glucose in it, which would be a closer environment to the tsetse gut.

2) It is not clear which parasite stage was used for tsetse infections, although everything indicates it was procyclics, in which case I don't think is the right one for these experiments. For the purpose of investigating the role SoMo during the early stages of a trypanosome infection in the tsetse, the authors should have used instead BSF. In fact, it'd be cleaner to use a tet-inducible conditional null instead of add back mutants, as sometimes add back clones differs in infectivity. The time in culture of PCF is critical for tsetse infectivity and tissue tropism, even if these parasites have been in culture only for few days.

3) In general, the quality of the fluorescent microscopy figures is not that great (e.g. Figure 7). The authors need to use other dyes and tools to better define the different fly tissues. The same happens with the videos of infection by WT parasites which, in some cases, it is difficult to know what tissues or what aspects of the tissues the authors are pointing out.

4) Importantly and in relation to the previous comment, why the authors did not include videos of PDEB1 mutants in the fly? To me, this is absolutely essential to demonstrate their point. If trypanosome do not cross the PM, where do they go? Do they get stuck in the PM or are they expelled in the frass?

5) It seems that only one biological replicate was made per experiment and using very low tsetse numbers. These experiments need more biological replicates to determine its significance.

6) The number of GPEET+ cells increases in KO cell lines at 7dpi (Fig 5C). Although GPEET expression may have nothing to do with SoMo (but may indicate that the cells are still as "early" procyclics), this delayed expression may also suggest a potential delay in PV colonization by PDEB1 mutants. Therefore, the PVs from flies infected with mutant trypanosomes should have been scored also at later time points to make sure that infection of this organ does not occur at a slower pace due to an overall delayed differentiation process in this KO cells.

Other comments:

1) Figure one is both wrong and low quality. First, the tissues used look broken; i.e. the midgut in panel B has a discontinues nuclei staining whereas the proventriculus shows a strange morphology and the nuclei staining is not defined as shown for the midgut section and looks like autofluorescence. In addition, in panel A, the proventriculus is misplaced and looks connected to the midgut by a fine thread, which I believe represents the anterior midgut.

2) Why using different statistical tests to determine significance in Figure 3A and 3B?

3) Why does the ectopic PDEB1 shows a lower apparent molecular weight with the same probe? Also, in Figure 4A, the second band on the left I believe is PDEB2 and not PDEB1 (as shown in Figure 2A).

4) A recent report (Schuster S. et al 2017, e-Life) showed evidence of "collective motion" occurring at different times of a trypanosome infection and in different tsetse tissues. Despite its potential importance to explain trypanosome migration within the tsetse, this publication was completely ignored by the authors.

Point by point response to reviewers:

Reviewer #1 (Remarks to the Author):

This is a nice attempt to relate some in vitro observations to the biology of the Tsetse lifecycle of the trypanosome parasite. It's important that people are starting to generate mutants and seeing how they behave in in vivo environments to try and tease out phenotypes not seen in culture. The Given that direct observations of the biochemistry in the infected fly is so difficult the conclusions depend upon a set of correlations between different in vitro and in vivo characteristics and measurements (motility and biochemistry in vitro and infectivity of gut areas in the in vivo). These correlations need to be very firmly based to rule out other possibilities in order that causal conclusions can be drawn. This is nice work but needs a little more to rule out some other possibilities – namely a cryptic motility defect or a differentiation (early / late procyclic) defect rather than a social motility cause.

Therefore I do have some reservations-

1. The title needs changing – the work here is not about navigation through tissues.

We offer an alternative title:

Flagellar cAMP signaling controls trypanosome progression through host tissues.

Please also note that there has been a change to an author's name: Sebastian Millius is now Sebastian Shaw.

2. Given the work on genetics of motility phenotype defects in other systems such as Chlamydomonas a rigorous assessment of motility is critical for the statement to be made that this is about flagellar signalling and not a motility defect. This is too important a result that the only data presented on the mutant movement is the graph of mean square displacement. They have the videos so lone would also expect to see the traces of the cell movement analysed in terms of directional issues, processivity, etc. It is perfectly possible for the MSDS graphs to be similar but the mutant cells to exhibit other motility phenotypes. Also, the standard assessment of motility should also include an analysis of high speed movies of the cells to show that the flagella behave in the same way. The results here can still be explained by some subtle motility defect.

We value the reviewer's feedback and agree that it is important to analyze motility carefully. We had originally performed MSD analysis because this takes processivity into account (for example, a cell moving only straight will have a greater MSD than a cell that curves or turns, even if they move at the same speed). We recognize the reviewer's concern however, and have therefore completed additional analyses as requested and have been very thorough (including additional MSD analyses, motility traces and high-speed movies). For example, in the original analysis we used fluorescent microscopy to track cells expressing fluorescent protein. In our experience UV exposure can alter motility although this should apply equally to the WT and KO. Nevertheless, to rule out any potential influence, we therefore performed our new analyses using dark field illumination on cells without fluorescent protein. We also increased the frame rate from 8.5 frames per second to 30 frames per second and tracked 1200 – 1400 cells from three independent biological replicates for each line. As a control for a known motility mutant, we also generated a trypanin knockout mutant (TPN KO) in the same background as the PDEB1-KO. Loss of trypanin results in cells that have limited processivity [Hutchings et al. 2002 J Cell Biol. vol. 156, p. 867-877]. Results are shown in Figures 2c, S2, movies 1-3.

As shown in Figure 2c, loss of PDEB1 does not result in loss of motility. In fact PDEB1-KO cells have a greater MSD than WT. We are not entirely sure of the reason for the difference between this analysis and the previous one done with UV-irradiated cells, but since these are less perturbing we report these analyses here. Moreover, in either case, there was no loss of motility for PDEB1-KO mutants. TPN KO cells on the other hand show almost no MSD compared to WT and PDEB1 KO, as expected (Figure 2c). In motility traces, no clear difference is evident for PDEB1-KO vs WT, while reduced processivity is clearly evident for the TPN-KO. Thus WT and PDEB1-KO cells are fully able to translocate, while TPN-KO cells are not. These results support our conclusion that the fly infection defect is not due to a loss of motility.

As noted above, MSD encompasses processivity. Therefore, increased MSD could indicate cells are moving faster and/or turning less. To assess this, we examined curvilinear velocity (VCL) and straight-line velocity (VSL) [Mortimer, 2000. J Androl. vol. 21, p. 515-524.] (Supplemental Figure S2). VCL represents the total distance that the cell travelled over time, and VSL represents the straight-line distance the cell moved from its initial position to its final position over time [Mortimer, 2000. J Androl. vol. 21, p. 515-524.]. Thus, plotting VCL vs VSL provides a measure of how linear or curving a cell's trajectory is. The VCL and VSL distribution of cell traces from Figure 2(b-d) show that WT and PDEB1 KO cells move similarly, while the trypanin motility mutant clearly shows reduced straight line velocity (Figure S2), reflecting their defect in processive motility. The ratio of VSL/VCL gives mean linearity for each population [Mortimer, 1990. J Androl. vol. 11, p. 195-203]. As shown in Supplemental Figure S2, the mean linearity of PDEB1 KO is similar though a bit larger than WT, while mean linearity of TPN KO cells is substantially reduced. These results indicate that the greater MSD in PDEB1-KO cells may reflect less turning compared to WT. It is notable that a similar phenotype, less turning, is observed for bacterial chemotaxis mutants [Parkinson and Houts, 1982. J. Bacteriology. vol.151, p.106-113].

Finally, as requested we've done high-speed (496 frames/second) video analysis of parasite motility (Figure 2d, movies 1-3). *T. brucei* propulsion is driven by a three-dimensional (3D) waveform that propagates from the tip of the flagellum toward the base, driving cell movement with the flagellum tip leading and the cell rotating on its long axis. Cells occasionally pause and may reorient [Branche et al., J. Cell Sci. 2006; Heddergott et al., PLoS Pathog. 2012]. In high speed video analyses, these fundamental features do not appear different for PDEB1-KO vs WT. While we cannot rule out a possibility that there may be minor differences that are not resolved in our analysis, both WT and PDEB1-KO cells exhibit a 3D, tip-to-base waveform, driving the cell forward with the flagellum tip leading and

the cell rotating. (Figure 2d, Movies 1-3) In contrast, the flagellum beat of TPN KO cells does not propagate regularly, resulting in a lack of cell translocation.

Overall these data reveal that PDEB1 KO cells do not show reduced motility and have no obvious change in flagellum waveform features. In fact, PDEB1 KO mutants translocate more than WT, perhaps reflecting the elevated cAMP in the cell [Oberholzer et al. 2015. mBio vol. 6, e01954-14], which would be expected to reduce its capacity to alter flagellum beating in response to fluctuating cAMP levels in the cell. As noted above, reduced turning under standard motility conditions is a feature of bacterial chemotaxis mutants. Thus, the combined results support the idea that the fly infection defect of PDEB1-KO mutants stems from improper regulation of motility, e.g. a signaling defect, rather than disruption of motility per se. The text has been modified on lines 158-174 to reflect these new experimental data.

3. The lines 176 – 182 suggest that the worry is that the KO cells have become late procyclic forms since the Social Motility phenotype is restricted to early forms. The authors suggest this is not an issue since the KO express the GPEET marker. However, does this not raise the more fundamental issue of whether the mutant cells in the tsetse (albeit fewer) that are able to establish the late infections are indeed late procyclics. Is that the mutants are defective in transition to the late forms in the fly and so there are less of them because of replication of this form not because of “navigation”? The type of parasites seen in the infections of both mutant and wild type surely need assessing because of this issue?

The reviewer may have overlooked that we analyze the ability of trypanosomes to down-regulate GPEET in the fly (text lines 224-233 in this version and Figure 5c and d). This analysis showed that the wild-type, knockout and addbacks differentiated with similar kinetics and were all GPEET-negative by day 14. We have changed the depiction in the graph (Figure 5c) to make it easier to understand and added a comment to the discussion (lines 349-351).

4. There is variability between the infection rates on different experiments (3a vs 5a), which is concerning with a significant infectivity effect seen in 3a but not 5a. Is there a general loss of fitness in the KO or not, the results make it difficult to be sure? Moreover, the use of t-tests seems inappropriate for analysing changes in categories a chi squared test would be better. The proventriculus effect does appear to be consistent between the experiments though.

As noted in the original text (lines 216-218 and 270-272 in this version), we agree with the reviewer that midgut infection rates were variable between experiments using independent batches of flies, while the proventriculus infection defect is unequivocal and consistent across 3 completely independent batches of flies and using independent cultures of each parasite line. It is for this reason that the message is that PDEB1-KO mutants are defective in proventriculus infection. Note that these experiments employ complex biological systems - live flies infected with live parasites - so some variability can be expected between different batches of flies, as has been reported by other labs. Crucially however, as noted by the reviewer, there is a marked defect in the KO's ability to infect the proventriculus and this is the key message from these experiments.

In answer to the question about fitness, we do not see a general loss of fitness in the KO (see growth curves in Figure 2 and Supplemental Figure S1).

As requested we re-analyzed the data using Fisher's exact test (which is appropriate for these numbers). The statistical significance remains the same across the board. The figures have been updated accordingly.

5. The use of a northern blot to confirm the KO seems odd as this just shows lack of expression not loss of the gene - a Southern or PCR test is needed to show definitively that the gene is gone and not that its expression has stopped. In that phase (but possibly not others or not in vivo) of the life cycle.

We apologize for not making it abundantly clear that we also did PCR to verify the replacement of PDEB1 by selectable markers. An example is shown below, and the text of the manuscript (lines 147-148) indicate this. The northern shows, in addition to loss of PDEB1 in the KO, that PDEB2 is not up-regulated as a compensatory mechanism.

6. I was confused by the timings in the discussion - early procyclic forms according to their data are lost by day 7 (fig. 5c - GPEET expression) yet on line 189 day 7 is when the trypanosomes cross the PM and enter the ectoperitrophic space - if this is the time that SOMO is required for movement across the PM why are they switching away from expression of GPEET to late procyclic markers and to a cell type not capable of SOMO?

To clarify, on day 7, a substantial proportion of the cells in this experiment (>20%) are still GPEET-positive. Also the time at which they cross the PM is an approximation. It is also not known whether differentiation to late procyclic forms takes place in the lumen, during crossing of the peritrophic matrix or in the ectoperitrophic space. We wrote that "day 7 is the approximate time when the parasites enter the ectoperitrophic space". In some other reports, for example according to Gibson et al., they get there after day 3 and are there by day 6. In an effort to clarify this we have amended the text to read "Between day 3 and day 7, trypanosomes cross the PM and enter the ectoperitrophic space [Gibson and Bailey, Kinetoplastid Biol Dis, 2003; Schuster et al., Elife 2017]. As mentioned in our answer to point 3, we have changed the format of this figure to make it easier to understand.

7. Reading the literature about early and late parasites it appears that glycerol is required to maintain early PCFs yet here this stock of 427 cells are maintained as early PCFs without glycerol, which seems counterintuitive?

The reviewer is correct that in the original publication [Vassella et al., Genes Dev 2000] we showed that Antat1.1 required glycerol to maintain the cells as early procyclic forms following differentiation from bloodstream forms. However, the Bern stock of Lister 427, which was used in this study and several others, stays >98% GPEET-positive (i.e. early) in the absence of glycerol [Ruepp et al., J. Cell Biol., 1997; Imhof et al., Euk. Cell 2015]. We also verified this directly in this manuscript (Supplemental Figure S1).

8. Given the authors earlier work and that of others on genetics of establishment of infections in the salivary gland how do the authors square the idea of mass movements via SoMo with the idea that there are large bottlenecks and a founder effect.

The bottleneck and founder effects are seen in the salivary gland population [Oberle et al., PLoS Path 2010]. For wild-type parasites, there is no bottleneck during the establishment of a midgut infection [Oberle et al., PLoS Path 2010], nor during the transition from the midgut to the proventriculus [Peacock et al., Kinetoplastid Biol Dis 2007; Peacock et al., PLoS NTD 2012]. Our data are consistent with this. Taken together, this indicates the bottleneck occurs between the proventriculus and salivary glands. SoMo occurs in early procyclic forms, which correspond to a life-cycle stage present in the midgut and therefore not subjected to a bottleneck.

9. It would've been very valuable to consider the tsetse infection phenotype of true (paralysed) motility defects relative to the SoMo defect they show. One issue is the general baseline for motility of any kind in this life cycle transition process. What is the magnitude of the defect relative to a cell that can't swim? This is essentially a question about the biological significance of the result.

A requirement for forward motility in the fly was reported by Rotureau and coworkers [Cell Micro. 2014]. As shown in Figure 2 and Figure S2, PDEB1-KO cells do not have a block in forward motility, so comparing to a cell that can't swim would not be an appropriate comparison. Notably, the DNAI1 mutant used in Rotureau et al. also has a growth defect *in vitro*, as do many motility mutants [Baron, Ralston et al. J. Cell Sci. 2007], making it hard to distinguish the impact of reduced forward motility versus the impact of reduced growth rate. Thus, while a detailed analysis of the requirement for forward motility in tsetse infection is of interest, doing so is beyond the scope of the current work and the relevance to our findings reported here is unclear.

Reviewer #2 (Remarks to the Author):

Trypanosoma brucei is transmitted by tsetse flies. Once inside the fly, the parasites have to overcome several challenges in order to survive and develop, including colonization of the midgut ectoperitrophic space. This step is essential for establishing an infection in the tsetse. How trypanosomes reach this compartment is still debatable, but the most accepted hypothesis is that they must cross the fly's peritrophic matrix (PM). In addition, it is well accepted that *in vitro* grown procyclic trypanosomes (the equivalent forms to the ones found in the tsetse gut) undergo social motility (SoMo). The molecular bases of trypanosomes SoMo are still unknown, but elegant work published by the same authors have shown that this phenomenon occurs only in early procyclic cells and that it may involve multiple pathways/genes, including cAMP signalling systems. In addition, whether SoMo is relevant for parasite colonization of the fly's gut has been a burning question within the trypanosome community. Thus, this is a relevant and important topic for understanding vector-parasite interactions in general.

In this MS the authors investigated the *in vivo* role of SoMo by infecting tsetse flies with a *T. brucei* phosphodiesterase PDEB1 deletion mutant, which shows a SoMo phenotype *in vitro*. By using live confocal microscopy analyses of labelled tsetse tissues and fluorescent parasites, the authors show evidence that PDEB1 mutants remained trapped in the tsetse gut lumen, suggesting a possible role in the migration to the ectoperitrophic space. However, although there is no doubt that this mutant shows an infection phenotype in the fly, the main problem of this paper is the bias interpretation that this is due to the lack of SoMo and not as direct consequence of having a defect in cAMP metabolism.

The reviewer may have missed that we addressed this point in the discussion (lines 388-389 of the original submission; lines 360 -361 in this version) - we explicitly stated that "While it is

premature to say whether the coordinated movement exhibited by trypanosomes during SoMo takes the same form in the insect vector” and we certainly did not intend to give a biased impression. It is worth noting that although additional work is needed to resolve this question, prior work has shown a correlation between SoMo defects in a different mutant and the inability to infect the tsetse [Imhof et al., Euk. Cell 2015]. We appreciate the reviewer’s feedback however, and we have rewritten a portion of the discussion to make this distinction clearer (lines 352-367).

Of course, defects in cAMP metabolism could lead to a SoMo phenotype, but so other other mutations that in principle are not metabolically related. Thus, given the lack of clarity and experimental evidence on the importance of trypanosome SoMo during colonization of the tsetse gut, the paper becomes merely descriptive and rather reports on the *in vivo* essentiality of another trypanosome gene.

We respectfully disagree with the suggestion that the work is merely descriptive. For one thing, the question of how parasites migrate through tsetse fly tissues has been raised several times and is a major question in the field. For example, to quote Dyer et al [2013. Trends Parasitol.]:

“How PF trypanosomes sense to migrate to either the ES or the PV is unknown, but it may have to do with the capacity to communicate and cooperate in response to an external signal (also known as social motility).”

Additionally, cAMP has been suggested to participate in parasite infection of the tsetse, but this idea has not been directly tested before our work. Thus, independently of whether cAMP is acting via SoMo here or via a separate pathway, our studies provide an important advance by providing *in vivo* data to support the requirement of cAMP signaling in fly infection. Moreover, we do that using innovative approaches by 1) employing a specific mutant, PDEB1, known to directly alter cAMP in live parasites; 2) generating addbacks to demonstrate requirement for the PDEB1 gene specifically; 3) using fluorescent parasites and fluorescent tissue imaging to delimit the precise anatomical location of the block in infection – note that our study is the first to combine simultaneous parasite and fly tissue imaging with mutant and phenotype analysis; 4) we report the first ever mutant to have been demonstrated to have this defect (lumen to EPS); 5) we demonstrate the infection defect is not due simply to a block in forward motility or defective parasite development; 6) Our combined work demonstrates that despite the fact that WT parasites have no trouble getting from the midgut to PV, this step of the infection is not a given, but depends on specific parasite features beyond just the physical capacity to move. Thus, our work provides several important insights into parasite transmission through the insect vector.

Below, I list several other concerns and suggestions that need to be considered by the authors in order to show a stronger evidence of the potential role of trypanosome SoMo in the tsetse:

1) The environment of the tsetse midgut (including the type and availability of some nutrients) is very different to the nutritional environment of any procyclics culture media. This means that the authors may have overlooked a growth phenotype of the PDEB1 mutants if these parasites are cultured in chemically defined media. I wonder if at least the authors have tested the growth phenotype of this mutant in media with no or little glucose in it, which would be a closer environment to the tsetse gut.

At the reviewer’s suggestion we compared the growth of wild-type and KO cells in SDM80, which can be made minus/plus glucose. Parasites were monitored over a period of 8 days. The presence or absence of glucose has no effect on the population doubling times for WT

versus PDEB1-KO cells. These data are now incorporated in the manuscript (lines 155-157) and Supplemental Figures S1h and S1i).

2) It is not clear which parasite stage was used for tsetse infections, although everything indicates it was procyclics, in which case I don't think is the right one for these experiments. For the purpose of investigating the role SoMo during the early stages of a trypanosome infection in the tsetse, the authors should have used instead BSF. In fact, it'd be cleaner to use a tet-inducible conditional null instead of add back mutants, as sometimes addback clones differs in infectivity. The time in culture of PCF is critical for tsetse infectivity and tissue tropism, even if these parasites have been in culture only for few days.

The life-cycle stage is mentioned in the experimental procedures (lines 447-449 in this version). We infected the flies with early procyclic forms, which is the life-cycle stage that is competent for SoMo. Infection with procyclic forms is accepted in the field, and there are publications from a number of labs in addition to ours (e.g. Brun lab: Schöni et al., *Z. Parasitenkd.* 1982; Bastin lab: Rotureau et al., *Cell Micro* 2014; Gibson lab: Peacock et al., *PNAS* 2011; PLoS Pathog. 2018). The lethal effect of elevated cAMP in BSF forms [Gould, *FEMS Micro Rev* 2011] also poses problems for use of BSF forms for these particular experiments. Importantly, some groups include additives to improve infection rates – their modes of action are unknown and we never do this.

We disagree with the reviewer that a tet-inducible conditional null would be cleaner. We have seen that conditional mutants can be leaky - this may not be obvious from Western blot analyses, but is apparent when the protein of interest has enzymatic activity. Importantly, we tested two independent addbacks which were derived from the PDEB1 KO and both regained the ability to progress to the proventriculus (Figure 5b). This is proof that time in culture is not the reason for the tissue tropism defect seen in the KO.

3) In general, the quality of the fluorescent microscopy figures is not that great (e.g. Figure 7). The authors need to use other dyes and tools to better define the different fly tissues. The same happens with the videos of infection by WT parasites which, in some cases, it is difficult to know what tissues or what aspects of the tissues the authors are pointing out.

The ability to visualize fly tissues and live parasites simultaneously has only been reported one time previously [Schuster et al., *Elife* 2017], and our study is the first time it has been done with a mutant and the first time that infections have been studied at this resolution. There are two methods to stain the peritrophic matrix, FITC beads and FITC wheatgerm agglutinin and we tested both of them. In our hands, wheatgerm agglutinin was superior. To aid in knowing what tissues are being shown, we also modified Movies (renumbered as Movies 7, 8 and 9 in this version) by adding arrows to label specific structures that are being pointed out and have adjusted the figure legends to clarify this. We also modified Figure 1 (see below).

4) Importantly and in relation to the previous comment, why the authors did not include videos of PDEB1 mutants in the fly? To me, this is absolutely essential to demonstrate their point. If trypanosome do not cross the PM, where do they go? Do they get stuck in the PM or are they expelled in the frass?

It appears the reviewer is asking if the parasites are stuck while crossing or are expelled with the frass. We see parasites in the midgut lumen, indicating that the population is not expelled in the frass (at least not all of it). For the question of whether the parasites get stuck in the PM, due to restrictions of imaging resolution, we do not know if PDEB1 parasites get stuck in the PM and better resolution than afforded by videos will be required. We aim to enhance resolution of imaging, but as noted above (Rev 2, point 3), simultaneous imaging of live mutant parasites and infected tsetse tissues is reported here for the first time ever, to

our knowledge. As such, further improvements on the method in order to address this specific question is beyond the scope of the current work. Note that Movie 4, which shows a mixed infection of PDEB1 KO-dsRed and untagged WT cells, shows a PDEB1-KO that is motile, consistent with our in vitro analysis that single cells of the mutant are capable of translocating.

5) It seems that only one biological replicate was made per experiment and using very low tsetse numbers. These experiments need more biological replicates to determine its significance.

For the key question, i.e. WT vs PDEB1-KO infection, the experiment was repeated three times, using three independent batches of flies and three independent cultures of each parasite line (two times using non-fluorescent parasites and once using DsRed-expressing parasites). For each of these three completely independent experiments, we used 53-81 infected flies per experiment for WT and 69-99 flies per experiment for the PDEB1-KO. (Total of 210 infections for WT and 245 infections for PDEB1-KO.) For PV infection the number is lower, because in two experiments we examined only those flies with heavily infected midguts. WT and PDEB1-KO were additionally tested in mixed infections. In all these experiments the PV infection defect is unequivocal and this defect was rescued in two additional sets of infections with two independent addback lines (149 infected flies). Therefore, we respectfully disagree with the suggestion that sufficient replicates were not performed.

6) The number of GPEET+ cells increases in KO cell lines at 7dpi (Fig 5C). Although GPEET expression may have nothing to do with SoMo (but may indicate that the cells are still as "early" procyclics), this delayed expression may also suggest a potential delay in PV colonization by PDEB1 mutants. Therefore, the PVs from flies infected with mutant trypanosomes should have been scored also at later time points to make sure that infection of this organ does not occur at a slower pace due to an overall delayed differentiation process in this KO cells.

At no point do we see an increase in the number of GPEET-positive cells. Although the KO and addback 1 have more positive cells on day 7 than WT and addback 2, by day 14 all four cell lines are essentially GPEET-negative. Both addbacks 1 and 2 colonize the proventriculus. In fact, there are no significant differences between the wild-type and the addbacks. The previous version of Figure 5c also appeared to cause difficulties for reviewer 1. In this revised version of the manuscript we have regrouped the data to make them clearer.

Other comments:

1) Figure one is both wrong and low quality. First, the tissues used look broken; i.e. the midgut in panel B has a discontinues nuclei staining whereas the proventriculus shows a strange morphology and the nuclei staining is not defined as shown for the midgut section and looks like autofluorescence. In addition, in panel A, the proventriculus is misplaced and looks connected to the midgut by a fine thread, which I believe represents the anterior midgut.

Part of the problem may be due to the fact that resolution may have been impacted by pdf generation. What appears to be discontinuous staining or broken tissue can occur due to invaginations or protrusions in gut tissue, placing part of the epithelia outside the focal plane. Nonetheless, at the reviewer's suggestion we have replaced the image in Figure 1 with an alternative example where the midgut is clearly intact.

In our hands the proventriculus does not autofluoresce and can look different depending on the plane in which it is imaged. This also has the result of bringing some nuclei into focus while others are not as well defined. This is shown by the following examples:

The proventriculus is not shown in Panel A. If the reviewer is referring to the drawing in Panel B, this is schematic, pointing out the relevant organs for this study. For simplicity it does not contain all anatomical details, such as the salivary glands or the connection to the crop.

2) Why using different statistical tests to determine significance in Figure 3A and 3B?

Thank you for pointing this out. See our reply to Reviewer 1, point 4.

3) Why does the ectopic PDEB1 shows a lower apparent molecular weight with the same probe? Also, in Figure 4A, the second band on the left I believe is PDEB2 and not PDEB1 (as shown in Figure 2A).

The endogenous copy of PDEB1 has a long 3' untranslated region (UTR) of 2.1 kilobases. The ectopic copy has a short 3' UTR of 200 bases. This results in the different sizes of the transcripts. The open reading frame (orf) is the same size in both and the blot was probed with the entire orf. We have amended the legend to clarify this point.

The reviewer is right that the second band was mislabeled. Thanks for pointing this out. It has now been corrected.

4) A recent report (Schuster S. et al 2017, e-Life) showed evidence of "collective motion" occurring at different times of a trypanosome infection and in different tsetse tissues. Despite its potential importance to explain trypanosome migration within the tsetse, this publication was completely ignored by the authors.

We respectfully point the reviewer to lines 82-84, 90, 192-194, 269, and 401 in the original version where we acknowledge the work of Schuster et al. We have added additional references to this work in the discussion (lines 361-364 in this version).

Reviewers' comments:

Reviewer #1 (Remarks to the Author):

The authors have added some new data. However, there are issues - some about the data needed to provide a firm conclusion in this (acknowledged) difficulty of showing a clear causal link to one phenomenon as an explanation for a general journal like Nature. The second issue is the balance of the writing that really does not balance out the possibilities of the various possibilities and conclusions.

The title is not a simple issue of replacing navigation with progression ...the issue is that the paper does not address movement through tissues. It deals with movement from one part of the insect gut to another. Also, not only that it is not the HOST that is being studied -- it is the VECTOR. Of course, one can take it that because there is an infective cycle of these parasites in the insect that it is a "host". However, to do so would be to go against all usage of the Host-Vector terminology in the protozoal pathogens. In addition a general reader would surely expect this paper to be about transfer of the pathogen from blood to brain in its host pathology aspect ...which is absolutely not what the paper is about.

The big thing is that as the authors acknowledge in the discussion is that you can't easily connect this inability to establish a proventriculus infection with a loss of SOMO...

There are a few of points that follow from this and from the new data (not in order of importance:

- 1) Given that they have done the PCR analysis of the KOs and shown it in the responses to the reviewers they should include that data in the supplementary manuscript and not have that as data not shown.
- 2) The authors say there is not a motility defect in the PDEB1 KO . That is right in one sense the KO cells are able to move in culture; however, the movement of KO is different from the parental cells and so in my opinion there is a change in motility that the authors have not acknowledged as an explanation for the phenotype.
- 3) As the authors have the addback cell lines and since the motility assays were requested, it would be good to run the in vitro motility analyses on these cells to see if they restore the movement to parental levels. Since not provided I recognise that this is yet more work but the judgment is about how to ensure clearer discussion and conclusion.
- 4) The focus on SOMO in the discussion is understandable but it does seem constantly odd that the possibility of this being a simple chemotactic response i.e. the KO is unable to respond to a cue from the host is reduced to the phrase 'and other activities that are necessary'. They allude to this earlier in the results when talking about the change in motility but not really in the discussion. Given that this is surely absolutely as plausible as the SOMO option I feel it should be discussed as well - chemotaxis has been postulated to be important in the related parasite Leishmania in the sand fly. One could discuss this trypanosome -vector system in the context of other protozoan parasite - vector interactions where movement, chemotaxis and differentiation triggers have been studied.

Reviewer #3 (Remarks to the Author):

The different comments and concerns of reviewer 2 are adequately addressed by the authors. The work presented in this manuscript does not show a direct significance of SoMo (observed behavior in cultured early procyclics) for the progress of the in vivo parasite development in the

tsetse midgut. This is now better and more unambiguously phrased by the authors in the discussion part. However, in the abstract a suggestive and 'biased' interpretation of the in vivo relevance of SoMo still remains. Therefore, the last sentence of the abstract should be re-phrased as follows: "These results show that cAMP signaling is crucial for successful transmission and correlates with the parasite's capacity to migrate through host tissues". The inclusion of SoMo here in this last sentence is still too suggestive for its in vivo relevance which cannot be justified from the experimental results presented in this study.

Point by point response to reviewer comments (author responses and alterations to text in blue).

Reviewer #1 (Remarks to the Author):

The authors have added some new data. However, there are issues - some about the data needed to provide a firm conclusion in this (acknowledged) difficulty of showing a clear causal link to one phenomenon as an explanation for a general journal like Nature. The second issue is the balance of the writing that really does not balance out the possibilities of the various possibilities and conclusions.

We think that we have been quite balanced in the writing throughout the manuscript, with a focus on two main messages: 1) we show that the peritrophic matrix is a barrier that has to be actively overcome, which was not known previously, and 2) we show that overcoming this barrier requires the parasite's flagellar cAMP signaling pathway. (These points are emphasized in the first two paragraphs of the Discussion section.) Nonetheless, we recognize that reviewer 1 retains concerns and in response, we've revised the discussion further, as indicated below in response to specific comments. We have also completed additional experiments requested, as indicated below in response to specific comments.

Regarding various possibilities and conclusions, we specify in lines 351-352 of the Discussion that the infection defect could arise from a defect in moving to the ectoperitrophic space, or in survival once there and we articulate our reasons for favoring a defect in movement. Following this, lines 362-366, we specify that movement entails physical movement and 'control' of movement, e.g. chemotaxis, and provide our reasons for favoring a defect in 'control' of movement as the primary reason for the PDEB1-KO defect. Thus, we think that we have provided a balanced consideration of possibilities and conclusions.

The title is not a simple issue of replacing navigation with progression ...the issue is that the paper does not address movement through tissues. It deals with movement from one part of the insect gut to another. Also, not only that it is not the HOST that is being studied -- it is the VECTOR. Of course, one can take it that because there is an infective cycle of these parasites in the insect that it is a "host". However, to do so would be to go against all usage of the Host-Vector terminology in the protozoal pathogens. In addition a general reader would surely expect this paper to be about transfer of the pathogen from blood to brain in its host pathology aspect ...which is absolutely not what the paper is about.

We wish to retain the title as it is for the reasons outlined below. We leave this decision up to the editor.

First, we respectfully disagree with the reviewer that using the word "host" for the fly is incorrect terminology. The tsetse fly is a vector, but by definition, it is also the definitive host of *Trypanosoma brucei*, since this is where sexual reproduction occurs, while mammals are intermediate hosts. The terminology that we use is employed in recent papers from other leading researchers in the tsetse field, for example the Engstler lab (Schuster et al., *elife*, 2017) Title "tsetse fly host", Abstract "here we introduce the host insect"; Gibson lab (Peacock et al., *PLoS Pathogens*, 2018) Abstract "mammalian and insect hosts"; Aksoy lab (Griffin et al., *BMC Microbiol.* 2018), Abstract "tsetse host".

Second, we deliberately made the title broadly appealing as befits the readership of Nature Communications. Reading the Abstract provides more detail. The word “tissues” is used to describe the peritrophic matrix material that is the interface between the gut lumen, ectoperitrophic space and proventriculus. This conveys that the parasite needs to move between different, non-contiguous compartments. Biological definitions of tissue also encompass intercellular material (in this case the PM).

The big thing is that as the authors acknowledge in the discussion is that you can't easily connect this inability to establish a proventriculus infection with a loss of SOMO...

Regarding specifically the proventricular defect, this defect is a consequence of the fact that the parasites cannot make the transition to the ectoperitrophic space in the first place. We recognize that paragraph 2 of the discussion in the prior version started with a sentence about the ‘proventriculus’ and this might have been confusing. We have reformulated this to avoid this confusion (lines 339-340).

Regarding whether one can easily connect the infection defect and SoMo, we point out that our main message is actually about connecting the infection defect and cAMP signaling. We’ve tried to be clear on this throughout the manuscript, e.g. with a focus on signaling in the Introduction and Discussion.

There are a few of points that follow from this and from the new data (not in order of importance:

1) Given that they have done the PCR analysis of the KOs and shown it in the responses to the reviewers they should include that data in the supplementary manuscript and not have that as data not shown.

These data have now been incorporated as Supplemental Figure 1.

2) The authors say there is not a motility defect in the PDEB1 KO . That is right in one sense the KO cells are able to move in culture; however, the movement of KO is different from the parental cells and so in my opinion there is a change in motility that the authors have not acknowledged as an explanation for the phenotype.

We did try to indicate this in the results and discussion of the prior text. To accommodate the reviewer’s concerns here, we have also revised the discussion to address this, lines 354-355, and lines 362-366.

3) As the authors have the addback cell lines and since the motility assays were requested, it would be good to run the in vitro motility analyses on these cells to see if they restore the movement to parental levels. Since not provided I recognise that this is yet more work but the judgment is about how to ensure clearer discussion and conclusion.

In response, we have completed motility analysis of the addback lines and results are now presented in the Results section (lines 207-208) and in Figures 4C and Supplementary Figure 3. As we observed for SoMo, the addbacks showed motility that was closer to the WT, although not exactly like it, and therefore support our conclusions.

4) The focus on SOMO in the discussion is understandable but it does seem constantly odd that the possibility of this being a simple chemotactic response i.e. the KO is unable to respond to a cue from

the host is reduced to the phrase 'and other activities that are necessary'. They allude to this earlier in the results when talking about the change in motility but not really in the discussion. Given that this is surely absolutely as plausible as the SOMO option I feel it should be discussed as well - chemotaxis has been postulated to be important in the related parasite Leishmania in the sand fly. One could discuss this trypanosome -vector system in the context of other protozoan parasite - vector interactions where movement, chemotaxis and differentiation triggers have been studied.

We agree that defective chemotaxis may explain the SoMo defect and the infection defect (although we would argue that considering chemotaxis to be simple is an understatement). We had tried to convey this point in the prior version of the manuscript, when we referred to bacterial chemotaxis mutants that showed reduced turning frequencies as seen for PDEB1 KO (lines 170 to 171 in the current version). The reviewer requests that we also comment on this in the discussion, which we agree is a good idea, and we have now done so (lines 362-366).

Note that we do not regard SoMo and chemotaxis as being mutually exclusive. In fact, as indicated in lines 125-131 of the Introduction, we consider SoMo to encompass movement and change in movement in response to external signals, i.e. chemotaxis. We've modified those lines in the Introduction to try to clarify and have specifically stated in the Discussion (lines 364-365) that SoMo entails both movement and chemotaxis, copied below for reference.

INTRODUCTION: "When early procyclic forms are cultured on a semi-solid surface, they exhibit a type of coordinated group movement termed social motility (SoMo) in which the parasites assemble into groups that sense signals from other cells and alter their movement in response^{36,37}. Genetic or pharmacological inhibition of PDEB1 blocks SoMo, while reduced expression or ablation of catalytic activity of specific ACs results in hypersocial behavior^{26,35}. These findings indicate that flux through the cAMP pathway regulates how the parasites respond to signals from their environment."

DISCUSSION: "These entail movement and the ability to respond to signals that alter the form or direction of motility, i.e. chemotaxis."

We've also revised line 427 regarding "...and other activities that are necessary" to avoid implying that we consider SoMo and chemotaxis as mutually exclusive.

Reviewer #3 (Remarks to the Author):

The different comments and concerns of reviewer 2 are adequately addressed by the authors. The work presented in this manuscript does not show a direct significance of SoMo (observed behavior in cultured early procyclics) for the progress of the in vivo parasite development in the tsetse midgut. This is now better and more unambiguously phrased by the authors in the discussion part. However, in the abstract a suggestive and 'biased' interpretation of the in vivo relevance of SoMo still remains. Therefore, the last sentence of the abstract should be re-phrased as follows: "These results show that cAMP signaling is crucial for successful transmission and correlates with the parasite's capacity to migrate through host tissues". The inclusion of SoMo here in this last sentence is still too suggestive for its in vivo relevance which cannot be justified from the experimental results presented in this study.

We have made the change suggested by the reviewer.